# FASTVGGT: FAST VISUAL GEOMETRY TRANSFORMER

**You Shen[1], Zhipeng Zhang[2], Yansong Qu[1], Xiawu Zheng[1], Jiayi Ji[1],**
**Shengchuan Zhang[1], Liujuan Cao[1]\***
[1]Key Laboratory of Multimedia Trusted Perception and Efficient Computing,
Ministry of Education of China, Xiamen University
[2]School of Artificial Intelligence, Shanghai Jiao Tong University

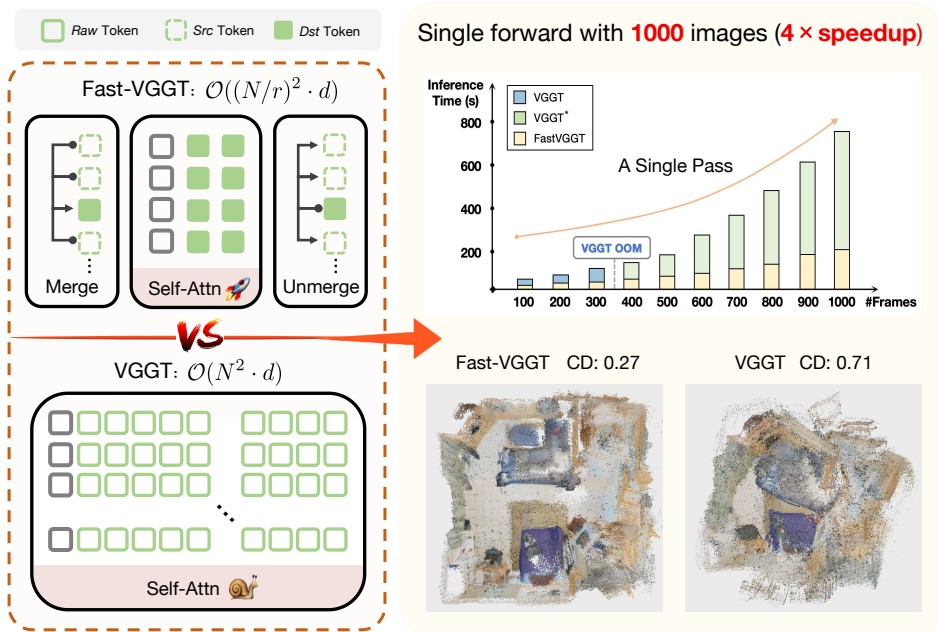

Figure 1: We propose FastVGGT, a training-free framework that processes 1,000 images in a single inference, achieving 4× faster while mitigating error accumulation. VGGT* refers to VRAM-Efficient of VGGT, enabling larger inputs.

## ABSTRACT

Scaling visual geometry transformers for long image sequences poses a significant computational and memory challenge. In this work, we diagnose this issue in the state-of-the-art model VGGT, and trace the primary bottleneck to its Global Attention layer. Our analysis reveals a "token collapse" phenomenon, where many tokens attend to nearly identical regions, resulting in redundant computation and inefficiency. Motivated by this finding, we propose FastVGGT, a training-free framework that strategically prunes these redundant tokens. Instead of uniform merging, FastVGGT employs a tailored, three-part token partitioning strategy. It preserves initial-frame tokens as a stable global reference, retains salient tokens to maintain fine details, and utilizes region-based random sampling to ensure spatially balanced coverage. Extensive experiments on multiple 3D geometry benchmarks validate our approach's effectiveness. Notably, on sequences of 1000 images, FastVGGT achieves a 4× speedup over the original VGGT while simultaneously mitigating error accumulation, demonstrating its efficiency and robustness for long-sequence scenarios. For further details, please visit our project page: https://mystorm16.github.io/fastvggt/.

---

\*Corresponding Author.

# 1 INTRODUCTION

Inferring the 3D geometric structure of a scene from visual inputs, is critical for enabling machines to understand and interact with the physical world. Recent advances in deep learning have catalyzed a paradigm shift in 3D geometric estimation (Wang et al., 2024b; Zhang et al., 2025; Qu et al., 2023; Wang et al., 2025a; Qu et al., 2025; Lu et al., 2025; Ye et al., 2024; Pan et al., 2024; Brachmann et al., 2024; Sun et al., 2021; Wang et al., 2025c; Lahoud et al., 2022; Zou et al., 2024; Fu et al., 2024), enabling a move from iterative, optimization-based pipelines to end-to-end neural networks that directly infer geometry from raw visual inputs. This transformation is exemplified by large-scale architectures like DUSt3R (Wang et al., 2024b) and its follow-ups (Yang et al., 2025; Tang et al., 2025; Leroy et al., 2024; Chen et al., 2024; Wang & Agapito, 2024), which showcase a remarkable capacity to reason about complex geometric relationships across image pairs.

Building upon this line of research, VGGT (Wang et al., 2025a) marks a significant advance. Its transformer-based, feed-forward architecture directly regresses key 3D attributes, including camera parameters, depth maps, and point tracks, to achieve highly stable and accurate reconstructions. While this establishes VGGT as a state-of-the-art framework for 3D scene understanding, its scalability is impeded by two critical bottlenecks. First, the model's reliance on dense global token interactions across views or frames results in prohibitive computational costs. Although used techniques like Flash-Attention mitigate the memory complexity from $O(n^2)$ to $O(nd)$, the underlying time complexity remains quadratic at $O(n^2d)$. Second, the global attention mechanism, essential for capturing inter-frame relations, is susceptible to error accumulation. As the token space expands with each new frame, minor inaccuracies are amplified, leading to significant prediction drift. Collectively, these limitations restrict VGGT's applicability in large-scale scenarios and motivate the development of more efficient and scalable architectures.

To pinpoint the primary inference bottlenecks in VGGT, we first conducted a detailed, component-wise performance analysis, as illustrated in Figure 2. The analysis reveals that while the computational costs of "Frame Attention" (intra-frame interaction) and "Global Attention" (cross-frame interaction) are comparable for short sequences, the cost of Global Attention escalates rapidly with sequence length, eventually dominating the entire runtime profile. This finding motivates our central research question: *Can the computational inefficiency of Global Attention be mitigated without compromising VGGT's capability?* To investigate this possibility, we visualized the attention maps in Figure. 3, which uncovered a crucial insight that attention patterns across tokens exhibit a high degree of similarity, indicating substantial redundancy in the global computation.

Motivated by the observation of attention redundancy, we adapt the training-free technique of token merging (Bolya et al., 2022; Bolya & Hoffman, 2023; Tran et al., 2024; Lee et al., 2024; Cao et al., 2023; Zeng et al., 2022; Renggli et al., 2022; Li et al., 2024; Choi et al., 2024; Feng & Zhang, 2023) to enhance VGGT's inference efficiency. Token merging consolidates redundant representations by partitioning tokens into source (src) and destination (dst) sets and merging each src token into its most similar dst counterpart. While effective in 2D vision tasks, its extension to structures designed

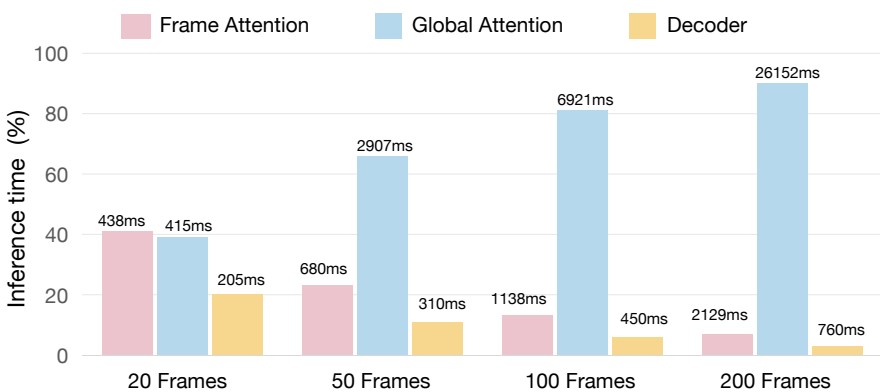

Figure 2: Component-wise analysis of VGGT inference time. As the number of input frames grows, the Global Attention module increasingly dominates the computational cost.

for 3D geometry understanding remains underexplored. Unlike 2D settings that process single images, VGGT relies on cross-image correspondences, making direct application of token merging highly challenging. To address this, we introduce FastVGGT, a novel, training-free framework that strategically applies token merging to mitigate the Global Attention bottleneck. Our approach begins by preserving the foundational coordinate system. Specifically, tokens from the initial frame, which serves as the global reference for the entire scene, are designated as high-priority destination (dst) tokens and are exempt from being merged to ensure reconstruction stability. Furthermore, to maintain global consistency and preserve fine-grained details, we identify and retain the most salient tokens across all frames, allowing them to bypass the merging process entirely and participate directly in the attention computation. Finally, inspired by ToMeSD (Bolya & Hoffman, 2023), we implement region-based random sampling within each subsequent frame. This ensures a spatially balanced selection of src and dst tokens, preventing critical information loss in localized regions during consolidation. It is worth emphasizing that although 2D token merging also relies on similarity cues, token collapse in VGGT reveals much higher redundancy, allowing the model to safely adopt much more aggressive merging ratios and achieve greater speedups. For instance, ToMeSD, which is designed for 2D generative tasks, degrades noticeably once the merging ratio exceeds 0.3, whereas FastVGGT maintains baseline-level performance even at a ratio of 0.9.

Our experiments demonstrate that this integrated approach allows FastVGGT to significantly reduce the computational overhead of Global Attention. For large-scale inputs of 1000 images, it achieves $4\times$ inference speedup over the baseline VGGT while simultaneously mitigating error accumulation in long-sequence reconstructions. Notably, the original VGGT suffers from prohibitive memory consumption, leading to Out-of-Memory ($OOM$) errors when processing sequences beyond 300 images. Through VRAM optimization, our modified VGGT successfully handles inputs of over 1000 images, demonstrating a substantial improvement in scalability.

In summary, our main contributions are as follows: 1) We identify and analyze the key bottleneck that limits VGGT's ability to process long sequences. 2) We uncover token collapse phenomenon in VGGT and introduce a specialized merging strategy that exploits this redundancy for acceleration. 3) Extensive experiments demonstrate that our method significantly accelerates VGGT on long sequences while preserving reconstruction quality. 4) For sequences exceeding 500 frames, our entirely training-free method surpasses the baseline by effectively mitigating error accumulation.

## 2 RELATED WORK

### 2.1 FEED-FORWARD 3D RECONSTRUCTION

Building on the foundations of traditional 3D reconstruction (Mur-Artal et al., 2015; Schonberger & Frahm, 2016; Mur-Artal & Tardós, 2017), recent end-to-end learning-based methods (Wu et al., 2023; Wang et al., 2024a; 2023; Zhang et al., 2024a; Chen et al., 2025; Zhang et al., 2024b; Fan et al., 2024; Lei et al., 2025; Smart et al., 2024; Wang et al., 2025b; Yu et al., 2024) leverage neural networks to encode scene priors, substantially enhancing robustness and cross-dataset generalization. Early progress was marked by DUSt3R (Wang et al., 2024b), which directly regresses view-consistent 3D point maps from only two RGB images without requiring camera calibration. The current state-of-the-art, VGGT (Wang et al., 2025a) scales this philosophy to a 1.2B-parameter transformer that jointly predicts camera intrinsics, extrinsics, dense depth, point maps, and 2D tracks. However, as input sequences grow longer, the global attention mechanism must capture inter-frame relations within an expanding token space. This not only increases computational overhead but also amplifies noise propagation, making long-sequence predictions more prone to drift. VGGT-Long (Deng et al., 2025) addresses the drifting issue by aligning sub-maps to suppress error accumulation, but at the cost of significantly reduced inference speed, undermining the efficiency of feed-forward 3D reconstruction. To overcome these challenges, we propose FastVGGT, which accelerates inference and mitigates error accumulation by reducing the number of tokens processed in Global Attention, thereby achieving a balance between efficiency and accuracy.

### 2.2 TOKEN MERGING

Visual token merging (Bolya et al., 2022; Renggli et al., 2022; Zeng et al., 2022; Haurum et al., 2023) was initially proposed as a training-free technique to increase the throughput of Vision Transformers

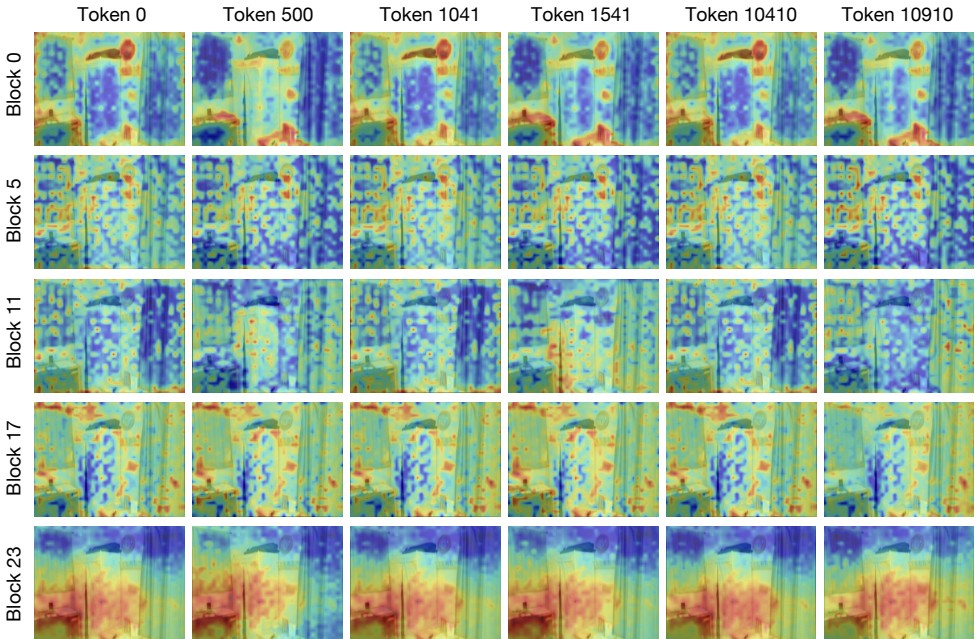

Figure 3: Visualizations of the Global Attention maps in VGGT, using six representative tokens (including the camera token and several image tokens), show that at every stage the attention patterns of different tokens exhibit a strong degree of similarity.

(ViTs) (Cao et al., 2023; Choi et al., 2024; Shen et al., 2024). These methods typically partition tokens into two groups, match each token with its nearest neighbor, merge them via average pooling, and concatenate the resulting set. Related efforts reduce token redundancy through adaptive selection, as in TokenLearner (Ryoo et al., 2021), modular superpixel tokens in A Spitting Image (Aasan et al., 2024), or clustering-based compression such as Agglomerative Token Clustering (Haurum et al., 2024). PiToMe Wang et al. (2025d) further proposes an energy–score–based criterion for deciding whether a region should be merged. Complementary to token-level compression, another line of work accelerates attention itself through kernel-based approximations. Nyströmformer (Xiong et al., 2021), inspired by the classical Nyström method (Williams & Seeger, 2000), approximates self-attention using landmark tokens, while Performer (Choromanski et al., 2020) employs random feature mappings for linear-time attention. Although simple and effective, most applications remain limited to the image domain, while long-form video remains underexplored despite its spatiotemporal tokens being both redundant and interdependent. Targeting feed-forward 3D reconstruction, we propose FastVGGT, which adapts token merging to VGGT and delivers significant acceleration without compromising reconstruction quality.

## 3 METHOD

### 3.1 VISUALIZATION ANALYSIS

We highlight an observation that motivates our optimization of the Global Attention module. As shown in Figure 3, we visualize VGGT's Global Attention maps on the ScanNet dataset. Each image is represented by 1,041 tokens (one camera token, four register tokens, and 1,036 patch tokens from a $28 \times 37$ grid). The dense self-attention mechanism generates an attention map for every token, and visualizations across tokens and blocks reveal that many of these maps are highly similar.

The phenomenon of attention similarity, often referred to as feature degradation, has been consistently reported in the DINO series (Oquab et al., 2023; Siméoni et al., 2025; Caron et al., 2021). In DINO, while the CLS token becomes increasingly discriminative, patch-level features gradually lose local consistency as they converge toward the CLS token, which undermines performance on dense prediction tasks. A similar limitation appears in VGGT: its global self-attention layers aggregate information across all tokens via weighted averaging, and without explicit regularization or task-specific constraints, the representational space is repeatedly compressed. This compression

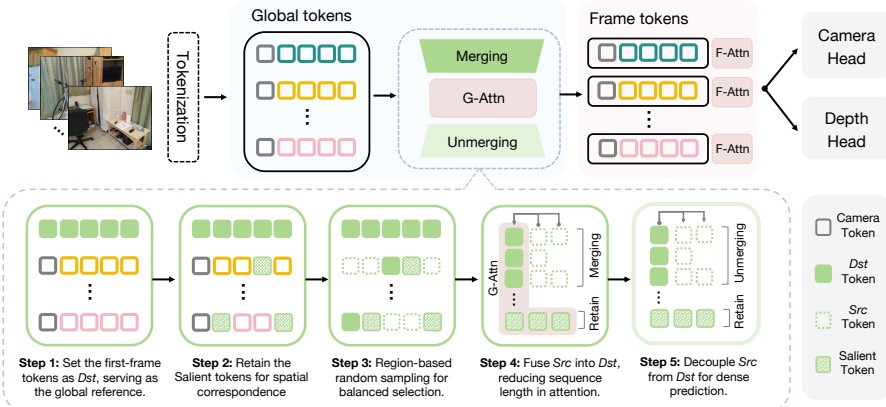

Figure 4: Overview of the proposed token merging strategy. The pipeline begins with tokenization of input frames. To alleviate the Global Attention bottleneck, we design a five-step procedure: (1) fix initial-frame tokens as destination (*Dst*) tokens, serving as the global reference for spatial consistency, (2) retain the top-k salient tokens to strengthen correspondences, (3) apply region-based random sampling for balanced selection, (4) fuse source (*Src*) tokens into their nearest *Dst* tokens during attention, and (5) decouple tokens via unmerging for dense reconstruction. G-Attn and F-Attn denote Global Attention and Frame Attention, respectively.

drives tokens to collapse along a dominant direction, eroding their distinctiveness. At a fundamental level, both DINO and VGGT lack explicit mechanisms to preserve token diversity, leading to a progressive loss of local variation.

It is important to note that the consequences of feature degradation differ markedly between DINO and VGGT. In DINO, the network must handle not only image classification but also dense prediction tasks such as segmentation, where local feature integrity is critical. As training progresses, patch tokens collapse toward the CLS token, weakening dense feature discrimination and directly impairing downstream performance. This motivates corrective strategies such as Gram Anchoring (Siméoni et al., 2025). By contrast, VGGT follows a two-stage design: Global Attention and Frame Attention. Global Attention enforces global consistency by capturing holistic spatial–temporal relationships. Here, the apparent degradation, in which tokens collapse toward a dominant subspace, can be interpreted not as a flaw but as a deliberate distillation of global semantics. Frame Attention subsequently reintroduces local variability, achieving a balance between global abstraction and local differentiation. At the same time, the strong feature similarity observed in Global Attention reveals redundancy that can be leveraged to reduce computational cost without compromising performance, thus offering a natural solution to VGGT's speed bottleneck.

Motivated by the strong similarities observed in Global Attention maps, we propose FastVGGT, which accelerates inference via token merging. Conventional strategies typically partition tokens into destination (*dst*) and source (*src*) sets through random or fixed-stride sampling. To assess their suitability, we tested both the random-sampling and fixed-stride variants of ToMe Bolya et al. (2022), as well as the more advanced PiToMe Wang et al. (2025d), which proposes an energy-score–based criterion for deciding whether a region should be merged. All methods were evaluated under three different merging ratios. As shown in Table 1, these approaches reduce inference time but severely degrade the Chamfer Distance, thereby harming reconstruction ac-

Table 1: Results of applying existing merging strategies. ToMe-S denotes the fixed-stride version of ToMe, and ToMe-R denotes the random-sampling version.

| Method | Ratio | 300 Frames | | 100 Frames | |
|---|---|---|---|---|---|
| | | CD ↓ | Time ↓ | CD ↓ | Time ↓ |
| ToMe-R | | 0.451 | 102.8s | 0.459 | 7.9s |
| ToMe-S | 0.3 | 0.633 | 98.4s | 0.548 | 7.9s |
| PiToMe | | 0.454 | 112.9s | 0.450 | 8.7s |
| ToMe-R | | 0.539 | 51.1s | 0.485 | 6.5s |
| ToMe-S | 0.6 | 0.563 | 49.3s | 0.523 | 6.4s |
| PiToMe | | 0.489 | 61.6s | 0.467 | 7.7s |
| ToMe-R | | 0.582 | 22.8s | 0.494 | 5.0s |
| ToMe-S | 0.9 | 0.708 | 21.2s | 0.628 | 4.8s |
| PiToMe | | 0.567 | 37.5s | 0.522 | 7.2s |
| VGGT | - | 0.416 | 131.4s | 0.423 | 9.1s |

curacy and failing to preserve VGGT's strong performance. This demonstrates that directly applying merging methods designed for 2D tasks to VGGT is non-trivial. To overcome this, we design three tailored strategies for feed-forward 3D reconstruction that preserves accuracy, accelerates inference, and mitigates error accumulation in long-sequence scenarios.

## 3.2 Token Partitioning

In contrast to token merging methods primarily developed for 2D vision tasks, which typically operate on individual images, feed-forward 3D reconstruction models process multi-view sequences with varying degrees of overlap across viewpoints. Such overlaps are critical for accurate 3D reconstruction, as they provide the cross-view correspondences necessary for recovering consistent geometry. Therefore, to achieve acceleration through merging while preserving reconstruction fidelity, it becomes essential to design an effective token partitioning strategy that operates across all frames in the sequence.

The design of our token partitioning strategy is governed by three principles: 1) preserving cross-frame correspondence to ensure structural consistency of the overall scene; 2) ensuring uniform merging within each frame to prevent both over-compression and redundancy; and 3) merging the most redundant tokens into the most representative ones to maximize acceleration efficiency. Accordingly, we divide tokens into three categories: salient tokens, which capture the most distinctive features of each frame; destination (dst) tokens, which act as representative anchors; and source (src) tokens, which correspond to redundant information to be merged. Based on these principles, we design three token partitioning strategies as detailed in the following.

**Reference Token Selection.** First, VGGT defines the first frame as the world coordinate system, with all tokens registered relative to this reference. This design makes the first frame a key anchor for maintaining spatial consistency across the sequence. Supplementary visualization of attention maps further shows that tokens consistently exhibit stronger activations toward the first frame than any other, highlighting its central role in guiding scene-level representations. Consequently, we designate all tokens from the first frame as dst tokens due to their strong representativeness.

**Salient Token Selection.** Second, since VGGT reconstructs scenes through cross-frame token interactions, a subset of key tokens is critical for establishing reliable correspondences across views. As illustrated in Figure 3, these tokens resemble distinctive keypoints in traditional matching algorithms. To preserve them, we extend conventional token merging by introducing a third category, dividing tokens into salient, dst, and src groups. Salient tokens are excluded from merging operations and instead participate directly in attention. For selecting salient tokens, we first apply a top-k strategy based on token norms to measure distinctiveness. However, its computational overhead grows with longer sequences. To improve efficiency, we adopt a fixed-stride sampling scheme that retains 10% of tokens per frame as salient tokens. Experiments show that this achieves accuracy comparable to top-k selection while greatly reducing cost. We therefore use fixed-stride sampling as the default strategy, balancing efficiency and effectiveness.

To further analyze the effects of the two strategies, we include additional experiments in Appendix E.2 that visualize the spatial distribution of salient tokens. We observe that top-K selection performs well in shallow blocks by accurately capturing semantically meaningful regions, but as the block depth increases, the selected tokens become increasingly concentrated. In contrast, although the fixed-stride strategy is less precise in locating key tokens, it consistently maintains a uniform spatial distribution across blocks.

**Uniform Token Sampling.** Finally, recognizing the dense prediction nature of 3D reconstruction, we ensure uniform intra-frame sampling to avoid local over-compression or redundancy. To this end, we assign dst and src tokens within each frame using a region-based random sampling strategy inspired by ToMeSD (Bolya & Hoffman, 2023) in diffusion models. Concretely, we first partition the input tokens by frame and arrange them into a 2D grid of image patches. Within each grid cell, dst tokens are sampled according to a predefined merging ratio with stride $K$, while the remaining tokens are designated as src tokens. This region-based strategy ensures spatially balanced merging and prevents artifacts such as the disappearance of large areas. Consequently, the merging process is more coherent, and the reconstructed scene better preserves global structural stability.

### 3.3 MERGING AND UNMERGING PROCEDURE

After token partitioning, *src* tokens are merged into their most similar counterparts in *dst*. Formally, given a token representation $x$, we compute cosine similarity between each $x_s \in src$ and all $x_d \in dst$ as $\text{sim}(x_s, x_d) = \frac{x_s \cdot x_d}{\|x_s\| \|x_d\|}$. Each source token $x_s$ is then assigned to its most similar destination token $x_d$, and updated by averaging $x'_d = \frac{x_d + x_s}{2}$. The updated $x'_d$ is retained while $x_s$ is temporarily discarded, thereby reducing the number of tokens processed in attention computations.

Dense 3D reconstruction requires per-token outputs. To satisfy this requirement, we adopt an unmerging operation, inspired by ToMeSD (Bolya & Hoffman, 2023), which restores the original token resolution and maintains full compatibility with the VGGT architecture. Specifically, suppose two tokens $x_1, x_2 \in \mathbb{R}^c$ are merged into a single representation $x^*_{1,2} = \frac{x_1 + x_2}{2}$. During unmerging, this representation is replicated to recover the original sequence length as $x'_1 = x^*_{1,2}$ and $x'_2 = x^*_{1,2}$.

### 3.4 VRAM-EFFICIENT IMPLEMENTATION

In our tests, the original VGGT encounters out-of-memory errors when processing sequences of around 300 frames. VGGT consists of 24 encoder blocks, but during inference only the outputs of layers 4, 11, 17, and 23 are required. Nevertheless, the original implementation stores intermediate results from all 24 blocks. To support longer input sequences, we introduce an simple optimized variant, VGGT*, which discards unused intermediate outputs during inference. This reduces memory consumption and enables processing of up to 1000 frames without affecting reconstruction quality. Figure 7 reports the GPU memory comparison between the original VGGT and VGGT*. Unless otherwise specified, all experiments in this paper use VGGT* as the baseline.

We clarify that although 2D token-merging methods reduce memory consumption by decreasing the number of attended tokens, this mechanism does not transfer effectively to VGGT. VGGT exhibits no meaningful token similarity within its frame-attention layers, and its alternating global–frame attention structure requires merged tokens to be fully restored before each frame-attention block when performing dense reconstruction, resulting in no reduction in the token count.

## 4 EXPERIMENTS

### 4.1 EXPERIMENTAL SETUP

We evaluate FastVGGT on three benchmark datasets: ScanNet, NRGBD, and 7 Scenes, with the best results highlighted in bold and the second-best results underlined. For ScanNet, which contains 1,500 scenes, we uniformly sample 50 scenes to form a reproducible benchmark, denoted as ScanNet-50. Our experiments focus on two tasks: camera pose estimation and point map reconstruction. Across both tasks, FastVGGT consistently achieves substantial speedups while maintaining accuracy. The overall architecture follows VGGT, comprising L=24 frame and global attention layers, with Flash-Attention2 (Dao, 2023) integrated to further accelerate inference. All input images follow VGGT's preprocessing, where the long side is resized to 518. To keep the method simple and efficient, we use a fixed merging ratio of 0.9 in Tables 2–7, which is sufficient to achieve substantial acceleration while maintaining performance comparable to the baseline. We further analyze the impact of different merging ratios in Table 8 and in the Appendix. All experiments are conducted on a workstation with an NVIDIA A800 GPU (80 GiB VRAM).

### 4.2 3D RECONSTRUCTION

**Comparisons on the ScanNet-50 Dataset.** We begin by evaluating FastVGGT on the ScanNet-50 dataset, reporting reconstruction quality using Chamfer Distance (CD). Experiments are conducted with input sequences of 1000, 500, and 100 images, enabling us to assess performance under varying sequence lengths. The results in Table 2 show that although SOTA methods such as $\pi^3$ and StreamVGGT achieve strong performance on short sequences, they fail on long sequences due to memory constraints. Methods like Fast3R (Yang et al., 2025) and CUT3R (Wang et al., 2025b) can process long sequences efficiently, but their reconstruction quality degrades severely. In contrast, FastVGGT delivers substantial acceleration over baseline VGGT across all settings while preserv-

Table 2: Quantitative results of point cloud reconstruction on the ScanNet-50 dataset with input sequences of 1000, 500, 300, and 100 images. *OOM* denotes out-of-memory.

| Method | 1000 | | 500 | | 300 | | 100 | |
|---|---|---|---|---|---|---|---|---|
| | CD ↓ | Time ↓ | CD ↓ | Time ↓ | CD ↓ | Time ↓ | CD ↓ | Time ↓ |
| $\pi^3$ (Wang et al., 2025d) | *OOM* | *OOM* | *OOM* | *OOM* | *OOM* | *OOM* | *OOM* | *OOM* |
| StreamVGGT (Zhuo et al., 2025) | *OOM* | *OOM* | *OOM* | *OOM* | *OOM* | *OOM* | *OOM* | *OOM* |
| Fast3R (Yang et al., 2025) | 0.684 | 397.8s | 0.701 | 97.3s | 0.711 | 34.9s | 0.723 | 4.8s |
| CUT3R (Wang et al., 2025b) | 0.786 | **34.8s** | 0.774 | **18.8s** | 0.775 | **11.1s** | 0.767 | **3.6s** |
| VGGT* (Wang et al., 2025a) | 0.471 | 724.6s | 0.420 | 177.5s | **0.416** | 131.4s | **0.423** | 9.1s |
| FastVGGT | **0.425** | 180.7s | **0.411** | 55.2s | **0.416** | 23.8s | 0.426 | 5.4s |

Table 3: Quantitative results of point cloud reconstruction on the 7 Scenes dataset. Stride denotes keyframes sampled every 3 or 10 frames.

| Method | 7 Scenes - Stride 3 | | | | | | | 7 Scenes - Stride 10 | | | | | | |
|---|---|---|---|---|---|---|---|---|---|---|---|---|---|---|
| | Acc ↓ | | Comp ↓ | | NC ↑ | | Time ↓ | Acc ↓ | | Comp ↓ | | NC ↑ | | Time ↓ |
| | Mean | Med. | Mean | Med. | Mean | Med. | | Mean | Med. | Mean | Med. | Mean | Med. | |
| $\pi^3$ (Wang et al., 2025d) | *OOM* | *OOM* | *OOM* | *OOM* | *OOM* | *OOM* | *OOM* | *OOM* | *OOM* | *OOM* | *OOM* | *OOM* | *OOM* | *OOM* |
| StreamVGGT (Zhuo et al., 2025) | *OOM* | *OOM* | *OOM* | *OOM* | *OOM* | *OOM* | *OOM* | *OOM* | *OOM* | *OOM* | *OOM* | *OOM* | *OOM* | *OOM* |
| Fast3R (Yang et al., 2025) | 0.045 | 0.027 | 0.047 | **0.010** | 0.616 | 0.627 | 43.7s | 0.040 | 0.021 | 0.056 | 0.013 | 0.639 | 0.657 | 5.5s |
| CUT3R (Wang et al., 2025b) | 0.179 | 0.121 | 0.097 | 0.043 | 0.588 | 0.581 | **14.5s** | 0.041 | 0.021 | 0.029 | **0.010** | 0.651 | 0.677 | **4.2s** |
| VGGT* (Wang et al., 2025a) | 0.019 | **0.008** | 0.027 | **0.010** | 0.611 | 0.628 | 76.7s | 0.020 | **0.008** | 0.027 | **0.010** | 0.623 | 0.641 | 8.7s |
| FastVGGT | **0.018** | **0.008** | **0.026** | **0.010** | **0.617** | **0.634** | 28.0s | **0.018** | **0.008** | **0.027** | **0.010** | 0.628 | 0.648 | 5.1s |

ing reconstruction accuracy. Notably, when processing very long sequences (*e.g.*, 1000 images), FastVGGT not only maintains reconstruction fidelity but also significantly mitigates error accumulation, demonstrating robustness and scalability for large-scale 3D reconstruction.

**Comparisons on 7 Scenes and NRGBD Datasets.** Following the CUT3R protocol, we evaluate FastVGGT on the 7 Scenes and NRGBD datasets. We report accuracy (Acc), completeness (Comp), and normal consistency (NC) using long-sequence inputs, with keyframes sampled every 3 or 10 frames. As shown in Table 3 and Table 4, FastVGGT maintains the robust performance demonstrated on ScanNet-50, further demonstrating its effectiveness for 3D reconstruction.

## 4.3 CAMERA POSE ESTIMATION

We evaluate FastVGGT on the ScanNet-50 dataset for camera pose estimation. Evaluation is conducted using four commonly adopted metrics: Absolute Trajectory Error (ATE), reflecting trajectory-level accuracy; Absolute Rotation Error (ARE), reflecting orientation accuracy; Relative Pose Error in rotation (RPE-rot), and Relative Pose Error in translation (RPE-trans), which jointly characterize local frame-to-frame consistency. As illustrated in Table 5, FastVGGT matches the baseline VGGT on shorter sequences (100 and 300 frames), while on longer sequences (500 and 1000 frames) it substantially reduces pose estimation error. Figure 5 further visualizes pose trajectories, demonstrating FastVGGT's ability to suppress cumulative drift, thereby underscoring its effectiveness and practicality for real-world deployment. Additional qualitative visualizations are provided in Appendix Figure 13.

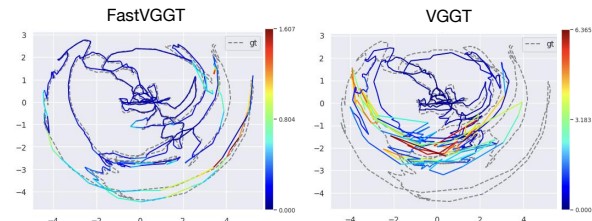

Figure 5: Comparison of pose estimation performance between FastVGGT and VGGT.

Table 4: Quantitative results of point cloud reconstruction on the NRGBD dataset.

| Method | NRGBD - Stride 3 | | | | | | | NRGBD - Stride 10 | | | | | | |
|---|---|---|---|---|---|---|---|---|---|---|---|---|---|---|
| | Acc ↓ | | Comp ↓ | | NC ↑ | | Time ↓ | Acc ↓ | | Comp ↓ | | NC ↑ | | Time ↓ |
| | Mean | Med. | Mean | Med. | Mean | Med. | | Mean | Med. | Mean | Med. | Mean | Med. | |
| $\pi^3$ (Wang et al., 2025d) | *OOM* | *OOM* | *OOM* | *OOM* | *OOM* | *OOM* | *OOM* | *OOM* | *OOM* | *OOM* | *OOM* | *OOM* | *OOM* | *OOM* |
| StreamVGGT (Zhuo et al., 2025) | *OOM* | *OOM* | *OOM* | *OOM* | *OOM* | *OOM* | *OOM* | *OOM* | *OOM* | *OOM* | *OOM* | *OOM* | *OOM* | *OOM* |
| Fast3R (Yang et al., 2025) | 0.074 | 0.036 | 0.024 | 0.011 | 0.658 | 0.682 | 68.9s | 0.061 | 0.028 | 0.031 | 0.013 | 0.669 | 0.712 | 7.4s |
| CUT3R (Wang et al., 2025b) | 0.346 | 0.243 | 0.184 | 0.090 | 0.579 | 0.623 | **18.3s** | 0.132 | 0.064 | 0.056 | 0.011 | 0.669 | 0.725 | **5.7s** |
| VGGT* (Wang et al., 2025a) | **0.029** | **0.019** | 0.018 | **0.009** | 0.727 | 0.728 | 136.1s | **0.016** | **0.010** | **0.017** | **0.009** | 0.735 | 0.738 | 13.9s |
| FastVGGT | **0.029** | 0.021 | **0.019** | 0.010 | **0.730** | **0.738** | 53.1s | 0.018 | 0.011 | 0.018 | **0.009** | **0.736** | **0.741** | 7.3s |

Table 5: Quantitative results of camera pose estimation on the ScanNet-50 dataset.

| Input | ATE ↓ | | ARE ↓ | | RPE-rot ↓ | | RPE-trans ↓ | |
|---|---|---|---|---|---|---|---|---|
| Frames | Baseline | Ours | Baseline | Ours | Baseline | Ours | Baseline | Ours |
| 1000 | 0.196 | **0.164** | 4.636 | **3.860** | 0.997 | **0.667** | 0.039 | **0.029** |
| 500 | 0.174 | **0.145** | 4.190 | **3.591** | 0.963 | **0.627** | 0.042 | **0.031** |
| 300 | 0.145 | **0.142** | 3.689 | **3.554** | **0.786** | 0.801 | 0.040 | **0.036** |
| 100 | **0.140** | 0.141 | 3.625 | **3.512** | **1.224** | 1.262 | **0.058** | 0.061 |

## 4.4 ABLATION STUDIES

**Token Partitioning.** We evaluate the effectiveness of different token partitioning strategies using 500-frame inputs from ScanNet-50. As shown in Table 7: (a) directly selecting dst and src tokens through random sampling yields poor performance; (b) adopting region-based intra-frame uniform sampling improves performance but remains suboptimal; (c) designating the global reference frame (first frame) as dst tokens brings substantial gains; and (d) further protecting salient tokens leads to the best performance.

**Location and Intensity of Merging.** To balance accuracy and efficiency, we evaluate the effect of varying the starting block for merging and the applied merging ratio. As shown in Table 8, increasing the merging ratio consistently reduces inference time, with only minor fluctuations in Chamfer Distance. Consequently, we adopt an aggressive strategy that applies a 90% merging ratio from block 0 across all subsequent layers, yielding a favorable balance of accuracy and efficiency.

**Salient token selection strategies.** In Table 6, we provide a clearer comparison between the topK–based and fixed-stride sampling strategies for selecting salient tokens. On ScanNet-50, we evaluate both performance and efficiency, and additionally analyze how different top percentages of tokens selected from the scene affect the results. The findings show that the fixed-stride strategy noticeably improves efficiency while maintaining performance comparable to the top-k approach. Considering both accuracy and efficiency, we ultimately adopt the fixed-stride strategy that samples 10% of the scene tokens.

## 4.5 QUANTITATIVE ANALYSIS OF TOKEN COLLAPSE

We conduct experiments in Figure 6 using average cosine similarity to analyze VGGT's token-collapse phenomenon. The experiments are performed on ScanNet-50, following the same setup as Figure 3. Specifically, we compute the similarity metric as the average pairwise cosine similarity across six tokens (0, 500, 1041, 1541, 10410, 10910) sampled from different frames. Cosine similarity ranges from -1 to 1, where 1 indicates highly similar token-attention patterns, 0 indicates uncorrelated patterns, and -1 indicates opposite patterns. The results reveal substantial similarity across most global-attention blocks, with noticeable drops occurring around Blocks 1 and 14.

Interestingly, even in layers where token similarity is relatively low, applying an aggressive merging ratio (0.9) does not noticeably degrade the final performance, as shown in Tables 2–5. To better understand this behavior, we refer to conclusions drawn from our attention-map visualizations:

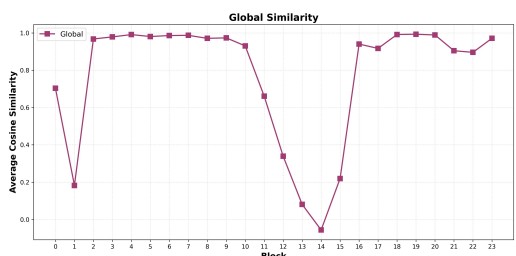
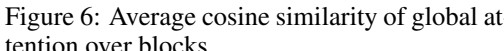
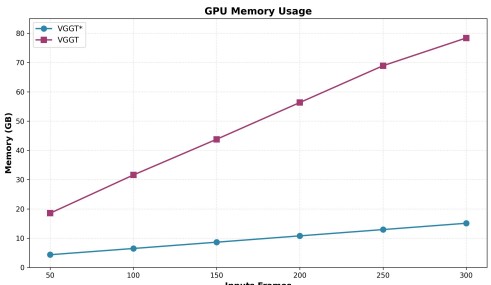

Figure 6: Average cosine similarity of global attention over blocks.

Figure 7: Comparison of GPU memory between the original VGGT and VGGT*.

Table 6: Quantitative results of salient token selection strategies on the ScanNet-50 dataset.

| Method | 500 | | 300 | | 100 | |
|---|---|---|---|---|---|---|
| | CD ↓ | Time ↓ | CD ↓ | Time ↓ | CD ↓ | Time ↓ |
| TopK-15% | 0.421 | 80.3s | **0.410** | 43.8s | 0.432 | 7.5s |
| TopK-10% | 0.423 | 73.8s | 0.423 | 35.3s | 0.428 | 6.7s |
| TopK-5% | 0.438 | 68.1s | 0.412 | 31.4s | 0.445 | 6.1s |
| Stride-15% | 0.430 | 62.8s | 0.429 | 28.5s | 0.434 | 6.0s |
| Stride-10% | **0.411** | 55.2s | 0.416 | 23.8s | **0.426** | 5.4s |
| Stride-5% | 0.425 | **52.5s** | 0.427 | **22.4s** | 0.438 | **4.9s** |

Table 7: Ablation for partitioning strategies.

| Methods | Uniform | Reference | Salient | CD ↓ | ATE ↓ |
|---|---|---|---|---|---|
| (a) | - | - | - | 0.947 | 0.842 |
| (b) | ✓ | - | - | 0.637 | 0.722 |
| (c) | ✓ | ✓ | - | 0.431 | 0.149 |
| (d) | ✓ | ✓ | ✓ | **0.411** | **0.141** |

Table 8: Ablation on ratio and blocks.

| Blocks | 0.3 | | 0.6 | | 0.9 | |
|---|---|---|---|---|---|---|
| | CD ↓ | Time ↓ | CD ↓ | Time ↓ | CD ↓ | Time ↓ |
| 0 | **0.408** | 119.8s | 0.415 | 64.3s | 0.411 | **55.2s** |
| 10 | 0.418 | 146.2s | 0.424 | 118.4s | 0.431 | 106.3s |
| 20 | 0.423 | 172.9s | 0.427 | 169.1s | 0.411 | 157.3s |

VGGT's global attention is fundamentally driven by feature matching, with high activation values concentrated around keypoint-like regions, as shown in Figures 11–12. Therefore, even though token similarity is lower in the middle blocks, we still apply an aggressive merging ratio, effectively retaining only a subset of the candidate matches. Our experiments show that this partial selection is already sufficient to maintain strong performance on long sequences. There is a conceptual parallel to classical SfM pipelines, where sparse yet reliable keypoints are often sufficient for high-quality pose estimation. Prior work (Zhao & Vela, 2020) has shown that increasing the number of matched keypoints does not necessarily improve performance; in fact, it can even degrade estimation quality by introducing additional outliers and noisy correspondences.

## 5 CONCLUSION

In this work, we propose FastVGGT, a training-free approach that accelerates VGGT inference through token merging while preserving reconstruction quality. Analysis identifies the global attention module as the main bottleneck for long-sequence inputs, and visualizations reveal a token collapse in attention maps, where tokens display strong similarity and redundancy. To address this, we propose three token partitioning strategies tailored to visual geometry tasks. Experiments show that FastVGGT achieves up to a 4× speedup on 1,000-image inputs with competitive accuracy in pose estimation and 3D reconstruction, while mitigating error accumulation in long sequences. Although extremely long sequences still have room for improvement before consistently outperforming shorter ones, FastVGGT provides both the practical capability and the conceptual foundation needed to make such progress feasible. We hope this work will inspire and guide future research on advancing long-sequence processing.

## 6 ACKNOWLEDGEMENTS

This work was supported by National Science and Technology Major Project (No. 2025YFE0113500), the National Science Fund for Distinguished Young Scholars (No. 62525605), and the National Natural Science Foundation of China (No. U25B2066, No. U22B2051, and No. 62272401).

## 7 ETHICS STATEMENT

This work focuses on improving computational efficiency in 3D scene reconstruction by introducing a training-free strategy. All experiments are conducted on publicly available benchmark datasets, including ScanNet, 7Scenes, and NRGBD. No human subjects, sensitive personal data, or privacy-related information are involved in this research. The proposed method does not present foreseeable risks of harmful misuse, and the study has no conflicts of interest or commercial sponsorship that could bias the results. We adhered to the ICLR Code of Ethics to ensure fairness, transparency, and academic integrity throughout the research process.

## 8 REPRODUCIBILITY STATEMENT

We have made extensive efforts to ensure the reproducibility of our results. The experimental setup, including datasets (ScanNet-50, 7Scenes, and NRGBD), input sequence lengths, evaluation metrics (e.g., Chamfer Distance, Absolute Trajectory Error, Relative Pose Error), and hardware environment (NVIDIA A800 GPU, 80GB VRAM), are clearly described in the main text. Comparative baselines (VGGT, Fast3R, CUT3R, etc.) and detailed ablation studies are reported to validate our claims. Supplementary materials will include code and inference scripts to enable researchers to replicate the experiments under the same conditions.

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

APPENDIX

# A   LLM USAGE AND REPRODUCIBILITY STATEMENT

The authors employed large language models (LLMs), such as Google's Gemini, solely to improve the language and readability of this manuscript. These models were not involved in generating ideas or shaping the conceptual framework. All authors have thoroughly reviewed the final version and take full responsibility for its content and claims. To ensure reproducibility, the source code of the core components has been submitted as anonymous supplementary material. Upon acceptance, all code and related resources will be released in a public GitHub repository.

# B   DISCUSSIONS OF TOKEN COLLAPSE PHENOMENON

## B.1   THE CONNECTION BETWEEN TOKEN COLLAPSE IN DINOV3 AND VGGT

We observe token collapse in both DINOv3 and VGGT, though its implications differ. In DINOv3, the multi-crop self-distillation framework drives the CLS token to capture view-invariant semantics by aligning local-crop student embeddings with global-crop teacher embeddings. This improves global consistency but gradually reduces patch-token diversity, as patch features increasingly converge toward the CLS token. Such dense feature degradation reflects an overemphasis on global invariance at the expense of local discriminability. To address this, Gram Anchoring constrains the similarity structure of patch features by matching the Gram matrix of the student to that of an early-stage teacher checkpoint with stronger dense properties. Given $P$ patches with $d$-dimensional $L_2$-normalized features $\mathbf{X}_S, \mathbf{X}_G \in \mathbb{R}^{P \times d}$ from the student and Gram teacher, respectively, the loss is defined as:

$$\mathcal{L}_{\text{Gram}} = \left\| \mathbf{X}_S \mathbf{X}_S^\top - \mathbf{X}_G \mathbf{X}_G^\top \right\|_F^2 . \tag{1}$$

By preserving patch-level similarity patterns while keeping the CLS representation robust, Gram Anchoring effectively alleviates token collapse in DINOv3.

In VGGT, collapse appears primarily in the global attention maps, where tokens exhibit strong similarity. Unlike DINOv3, this redundancy is less harmful, since global attention is designed to capture cross-view correspondences, echoing the classical role of "correspondence" in 3D vision. However, the redundancy inflates computation. Token merging provides an effective remedy by compressing redundant tokens while retaining reference and salient ones, thereby reducing inference cost and stabilizing long-sequence prediction.

Taken together, these observations indicate that token collapse is closely tied to architectural and training design choices—such as model structure, augmentation, and supervision—and becomes more pronounced in large models trained for long schedules. It is therefore a crucial factor when scaling both 2D representation learning and 3D geometry modeling.

# C   DISCUSSIONS OF SHORT SEQUENCES PERFORMANCE

Although long sequences can be effectively accelerated using an aggressive merging ratio while maintaining accuracy, we additionally conducted short-sequence experiments to provide a more comprehensive analysis of how the merging ratio interacts with sequence length.

Table 9 evaluates the impact of different merging ratios on ScanNet-50 with input lengths of 50, 30, 10, and 5 frames. The results show that aggressive ratios (e.g., 0.9) lead to noticeable performance degradation—and even failure cases, as illustrated in Figure 8, whereas a ratio of 0.3 achieves performance comparable to the baseline. These findings further indicate that the optimal merging ratio is highly dependent on the available temporal context, and that short sequences are substantially more sensitive to over-aggressive token reduction.

# D   DISCUSSION OF ROBUSTNESS TO THE FIRST FRAME

We include additional experiments evaluating robustness to first-frame blur and occlusion in Table 10, along with more extensive visualizations in Figure 9. On ScanNet-50, we apply a blur kernel

Table 9: Quantitative results of short sequence performance on the ScanNet-50 dataset.

| Input | Ratio | ATE ↓ | ARE ↓ | RPE-rot ↓ | RPE-trans ↓ | Time ↓ |
|---|---|---|---|---|---|---|
| 50 | 0.9 | 0.143 | 3.474 | 1.820 | 0.087 | **2.230s** |
| | 0.6 | 0.142 | **3.447** | 1.823 | 0.089 | 2.315s |
| | 0.3 | **0.137** | 3.466 | 1.698 | **0.079** | 2.859s |
| | Baseline | 0.138 | 3.512 | **1.685** | **0.079** | 3.478s |
| 30 | 0.9 | 0.172 | 3.750 | 2.826 | 0.150 | **1.348s** |
| | 0.6 | 0.166 | **3.679** | 2.722 | 0.146 | 1.373s |
| | 0.3 | 0.158 | 3.693 | 2.610 | 0.130 | 1.567s |
| | Baseline | **0.154** | 3.705 | **2.543** | **0.125** | 1.977s |
| 10 | 0.9 | 0.250 | 5.316 | 6.225 | 0.369 | **0.628s** |
| | 0.6 | 0.246 | 4.460 | 5.148 | 0.306 | 0.637s |
| | 0.3 | 0.201 | 4.326 | 5.003 | 0.255 | 0.651s |
| | Baseline | **0.197** | **4.131** | **4.683** | **0.247** | 0.712s |
| 5 | 0.9 | 0.452 | 18.626 | 22.989 | 0.682 | **0.433s** |
| | 0.6 | 0.452 | 17.905 | 22.179 | 0.673 | 0.440s |
| | 0.3 | **0.330** | 10.568 | 12.687 | **0.434** | 0.443s |
| | Baseline | 0.345 | **10.463** | **12.629** | 0.450 | 0.507s |

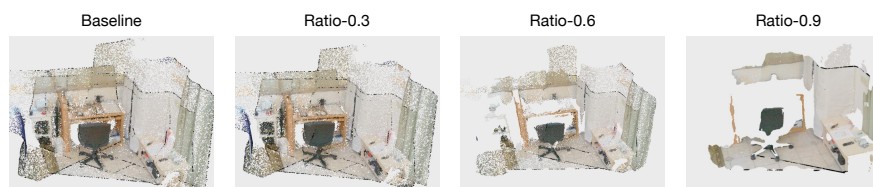

Figure 8: Comparison of merging ratios with 5 input frames, where aggressive ratios (e.g., 0.9) lead to failure cases.

of size 201 ($\sigma = 30$) and use a 40×40 mask that occludes either 50% or 80% of the first frame. Interestingly, FastVGGT achieves better performance than the baseline VGGT even under these challenging conditions. To further understand this behavior, we quantify the contribution of first-frame dst tokens: with 500 input frames, only 1.96% of merged dst tokens originate from the first frame, and with 100 input frames the proportion is 9.17%. These statistics indicate that when the first frame is degraded, the merging process naturally shifts its reliance toward higher-confidence matches from later frames, thereby mitigating the negative impact of early-frame corruption.

In addition, this phenomenon can be analogized to traditional SLAM systems, where the first frame is typically used for keypoint initialization. Corruption in this frame can compromise both the quantity and reliability of the extracted keypoints. When these degraded keypoints are matched against highly redundant ones from subsequent frames, the likelihood of mismatches and noise is significantly increased. In contrast, FastVGGT's merging strategy mitigates such issues by reducing redundancy, thereby helping to suppress the propagation of noise introduced by the first-frame.

# E  CORE CODE EXPLANATION

## E.1  PSEUDOCODE OF FASTVGGT

Algorithm 1 outlines the inference procedure of FastVGGT. The method begins by tokenizing each input frame into patch tokens, which are grouped into three categories: (*i*) reference tokens from the first frame, (*ii*) a fixed fraction of salient tokens that remain non-mergeable, and (*iii*) the remaining tokens that are assigned as mergeable sources or destinations via region-based sampling.

At each transformer block, source tokens are merged into their most similar destination tokens before global attention is applied, thereby reducing redundancy and computational cost. After the global attention step, merged tokens are restored by replicating the representations of their destinations,

Table 10: Quantitative results of first-frame occlusion or blur on the ScanNet-50 dataset.

| First Frame | Method | 500 | 300 | 100 |
|---|---|---|---|---|
| Occlusion-50% | VGGT | 0.501 | 0.482 | 0.475 |
| | FastVGGT | **0.471** | **0.452** | **0.441** |
| Occlusion-80% | VGGT | 0.514 | 0.503 | 0.488 |
| | FastVGGT | **0.476** | **0.468** | **0.459** |
| Blur | VGGT | 0.509 | 0.495 | 0.470 |
| | FastVGGT | **0.477** | **0.461** | **0.454** |

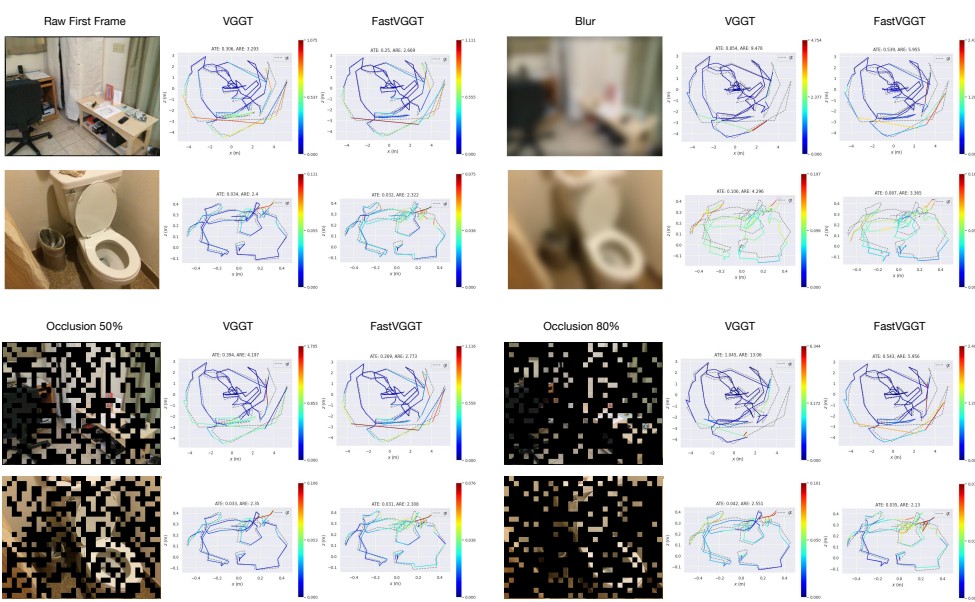

Figure 9: Comparison of FastVGGT and VGGT under first-frame occlusion or blurring.

ensuring compatibility with downstream decoding. Frame attention is then applied to all tokens without merging, preserving temporal consistency. Finally, the complete token set is decoded into dense 3D outputs, including depth maps, camera poses, and point maps.

### E.2 TOKEN PARTITIONING

We convert a global index buffer into ordered source ($a\_idx$) and destination ($b\_idx$) token indices, where destinations are predesignated ($-1$ in idx_buffer_seq). If protection is enabled, we additionally record protected_idx to exclude high-importance tokens from merging. The split function materializes these index sets on any tensor $x$, returning src/dst subsets (and protected if applicable).

```
rand_idx = idx_buffer_seq.reshape(1, -1, 1).argsort(dim=1)
num_dst_orig = int((idx_buffer_seq == -1).sum())

# Original src and dst indices
a_idx_orig = rand_idx[:, num_dst_orig:, :]
b_idx_orig = rand_idx[:, :num_dst_orig, :]
a_idx = a_idx_orig
b_idx = b_idx_orig

if enable_protection:
    protected_idx = protected_indices.unsqueeze(0).unsqueeze(-1)
```

---

**Algorithm 1** FastVGGT Inference (Training-Free Token Merging)

---

**Require:** Image sequence $\{I_f\}_{f=1}^F$, merging ratio $r$
**Ensure:** Dense 3D reconstruction outputs
 1: Tokenize each frame $I_f$ into tokens $T_f$; let $T \leftarrow \cup_f T_f$
 2: **Partition tokens:**
  (a) Set all tokens of frame 1 as destination $D$
  (b) Select $\rho$ fraction salient tokens $S_{\text{sal}}$ (fixed stride, non-mergeable)
  (c) Region-based sampling in each frame: assign remaining tokens to $D$ or $S$
 3: **for** each transformer block $\ell$ **do**
 4: **if** $\ell$ is Global Attention **then**
 5:  **Merge:** for each $x_s \in S$, find nearest $x_d \in D$ by cosine similarity; update $x_d \leftarrow (x_d + x_s)/2$
 6:  Run Global Attention on $D \cup S_{\text{sal}}$
 7:  **Unmerge:** replicate $x_d$ back to its merged $x_s$
 8: **else**
 9:  Run Frame Attention on all tokens
10: **end if**
11: **end for**
12: Decode tokens to outputs (depth, pose, point maps, etc.)

---

```
    num_protected = protected_idx.shape[1]
else:
    protected_idx = None
    num_protected = 0

num_src = a_idx.shape[1]
num_dst = b_idx.shape[1]

# Define an internal function to separate tokens
def split(x):
    C = x.shape[-1]

    if enable_protection:
        src = gather(x, dim=1, index=a_idx.expand(B, num_src, C))
        dst = gather(x, dim=1, index=b_idx.expand(B, num_dst, C))
        protected = gather(
            x, dim=1, protected_idx.expand(B, num_protected, C)
        )
        return src, dst, protected
    else:
        src = gather(x, dim=1, index=a_idx.expand(B, num_src, C))
        dst = gather(x, dim=1, index=b_idx.expand(B, num_dst, C))
        return src, dst
```

### E.3  MERGING

Given src→dst pairings, we preserve unmerged tokens ($unm$) and aggregate selected source tokens into their matched destination tokens via scatter_reduce along the sequence dimension, using a linear reducer (mean by default). The same indices are applied to any extra tensors (e.g., $q\_rope$, $k\_rope$, $v$) to ensure consistent downsampling across modalities.

```
# Extract unmerged src tokens - using actual unm_idx size
unm_len = unm_idx.shape[1]
unm = gather(src, dim=-2, index=unm_idx.expand(n, unm_len, c))
src_len = src_idx.shape[1]
src = gather(src, dim=-2, index=src_idx.expand(n, src_len, c))
dst = dst.scatter_reduce(-2, dst_idx.expand(n, src_len, c), src)
```

### E.4 Unmerging

To restore the pre-merge layout, we reconstruct a full-length tensor $out$ by scattering $dst$, $unm$, and reconstructed $src$ back to their original indices ($b\_idx$ for destinations; mapped $a\_idx$ entries for unmerged and merged sources). If protection was enabled, $protected$ tokens are also scattered back to preserve their original positions.

```
_, _, c = unm.shape
src = gather(dst, dim=-2, index=dst_idx.expand(B, src_len, c))
out = torch.zeros(B, N, c, device=x.device, dtype=x.dtype)
out.scatter_(dim=-2, index=b_idx.expand(B, num_dst, c), src=dst)
out.scatter_(
    dim=-2,
    index=gather(
        a_idx.expand(B, a_idx.shape[1], 1), dim=1, index=unm_idx
    ).expand(B, unm_len, c),
    src=unm,
)

out.scatter_(
    dim=-2,
    index=gather(
        a_idx.expand(B, a_idx.shape[1], 1), dim=1, index=src_idx
    ).expand(B, src_len, c),
    src=src,
)

if enable_protection:
    out.scatter_(
        dim=-2,
        index=protected_idx.expand(B, num_protected, c),
        src=protected,
    )
```

## F MORE VISUAL RESULTS

### F.1 Attention Map Visualization

Please refer to our project page https://fastvggt.github.io/ for more visualizations on the ScanNet dataset. To demonstrate the generality of our observations on the attention maps, we provide additional visualizations of token attention maps from the global attention layer of VGGT on the segment-102751 scene of the Waymo outdoor dataset. Figures 11 and Figures 12 show the results for the reference frame and the tenth frame, respectively. These visualizations allow us to examine the behavior of VGGT's global attention mechanism in greater depth, leading to the following conclusions:

1. Across both indoor and outdoor datasets, the attention maps corresponding to different tokens exhibit a consistently high degree of similarity, suggesting that the model captures common structural patterns regardless of scene type.

2. As the number of blocks increases, the overall distribution of the attention maps follows a pattern of concentration, dispersion, and re-concentration. We hypothesize that this pattern reflects the emergence of a more sophisticated mechanism for keypoint matching.

3. The similarity among attention maps becomes increasingly pronounced with greater block depth, indicating that deeper layers reinforce the alignment across frames.

4. In the final block, high activation values are predominantly concentrated on the reference frame and on frames that share a high degree of co-visibility with it, while frames with low co-visibility generally exhibit low activation values.

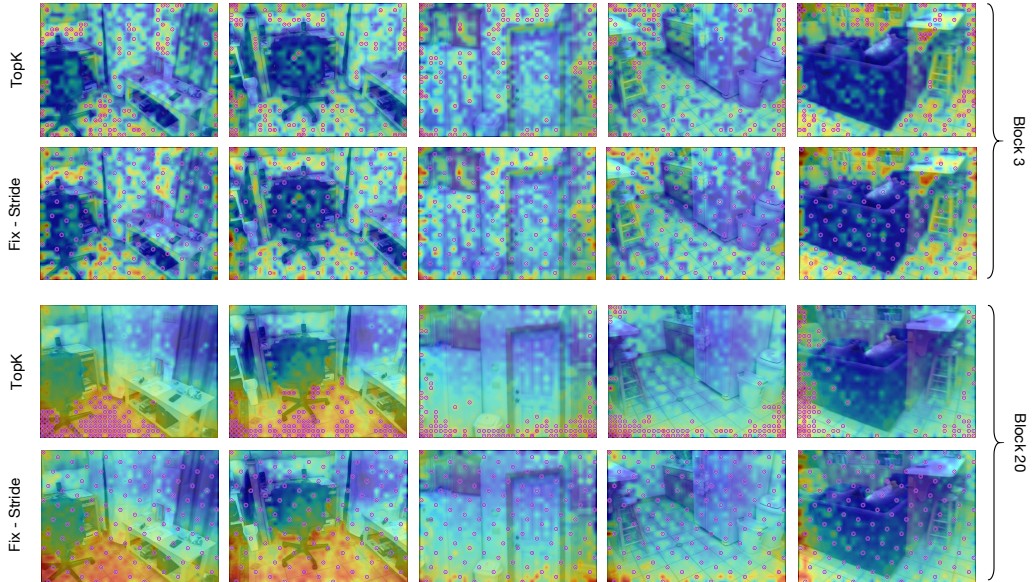

Figure 10: Visualization of different salient-token selection strategies on ScanNet-50.

### F.2 SALIENT TOKENS VISUALIZATION

To further analyze the effects of the two strategies, we visualize the spatial distribution of salient tokens in Figure 10. The results show that top-K performs well in shallow blocks by accurately capturing semantically important regions of the scene, but as the block depth increases, the selected salient tokens become increasingly concentrated. In contrast, although the fixed-stride strategy is less precise in identifying key token locations, it consistently distributes salient tokens more uniformly across different blocks.

## G  LIMITATIONS AND FUTURE WORKS

While FastVGGT demonstrates notable efficiency and robustness in long-sequence 3D reconstruction, several limitations remain:

- **Adaptability to sequence length.** The current design is tailored for large-scale inputs. When applied to short sequences, aggressive merging ratios may remove valuable fine-grained details and degrade performance. Developing adaptive merging policies that adjust to input size and scene complexity is therefore a promising direction.

- **Extension to other architectures.** Recent follow-up works of VGGT, such as PI3, also exhibit token redundancy patterns according to our preliminary observations. Extending FastVGGT to these architectures could further enhance their scalability and efficiency.

- **Long-horizon consistency.** Despite the strong generalization capabilities of both VGGT and FastVGGT, cumulative drift persists in very long sequences. Future research should focus on mechanisms that improve long-horizon consistency, potentially by combining token merging with temporal alignment or memory-based strategies.

- **Handling dynamic scenes.** The current framework does not explicitly address dynamic objects, which can lead to degraded reconstruction quality in highly dynamic environments. Future work may explore both data-driven solutions (e.g., incorporating motion-aware supervision) and architectural extensions (e.g., integrating dynamic object filtering or motion segmentation modules) to improve robustness in such scenarios.

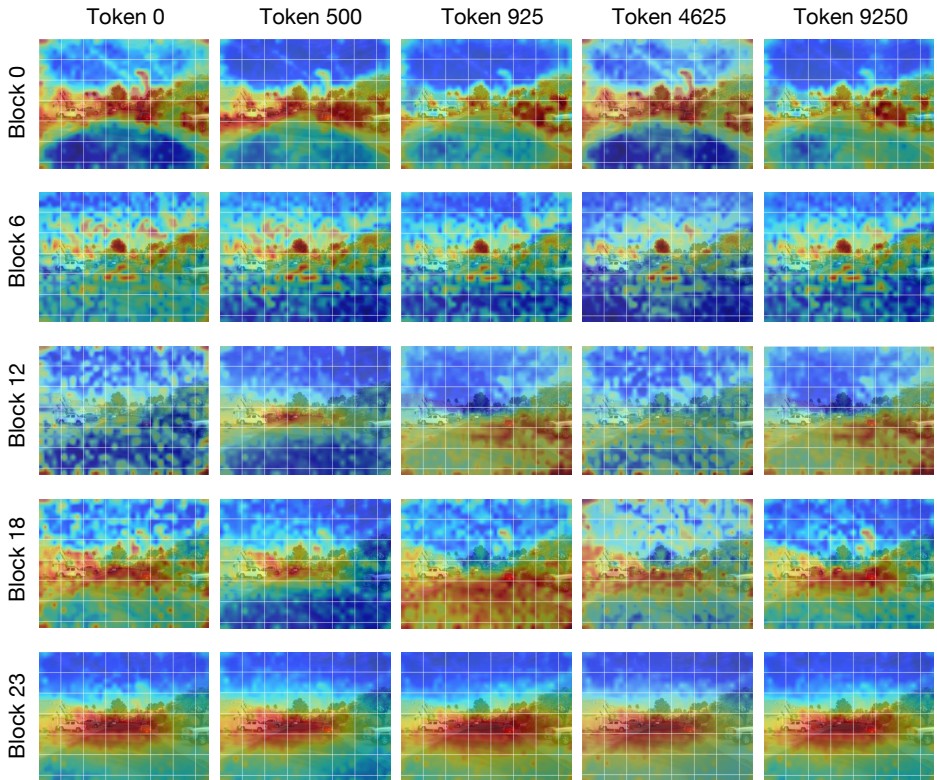

Figure 11: Visualization of attention maps from the global attention layer of VGGT for the reference frame in the segment-102751 scene of the Waymo dataset.

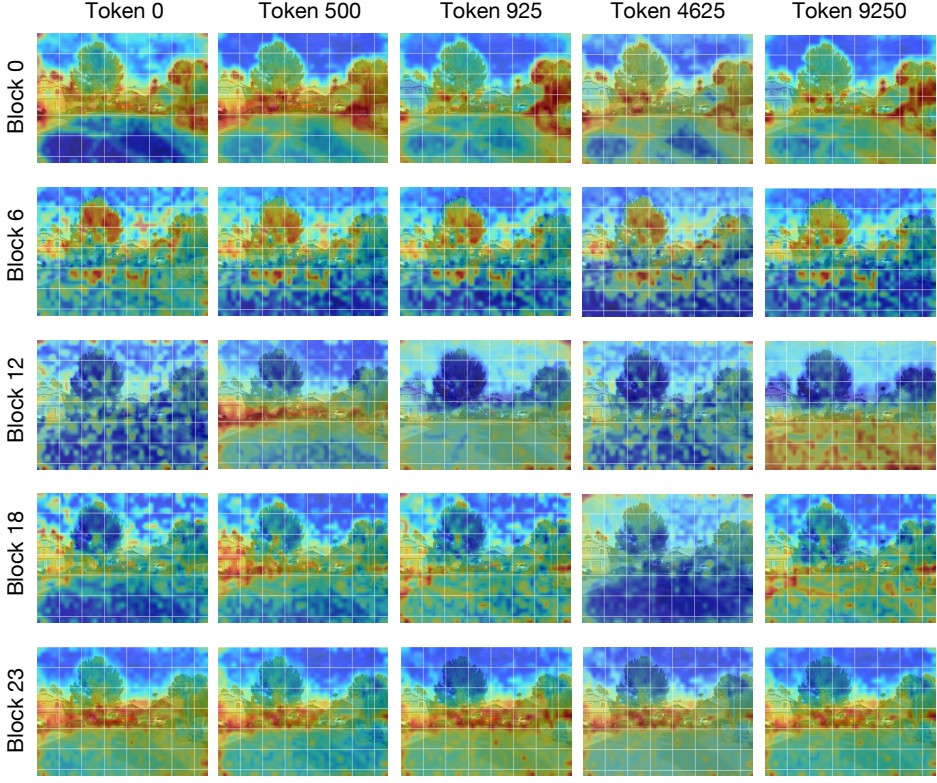

Figure 12: Visualization of attention maps from the global attention layer of VGGT for the tenth frame in the segment-102751 scene of the Waymo dataset.

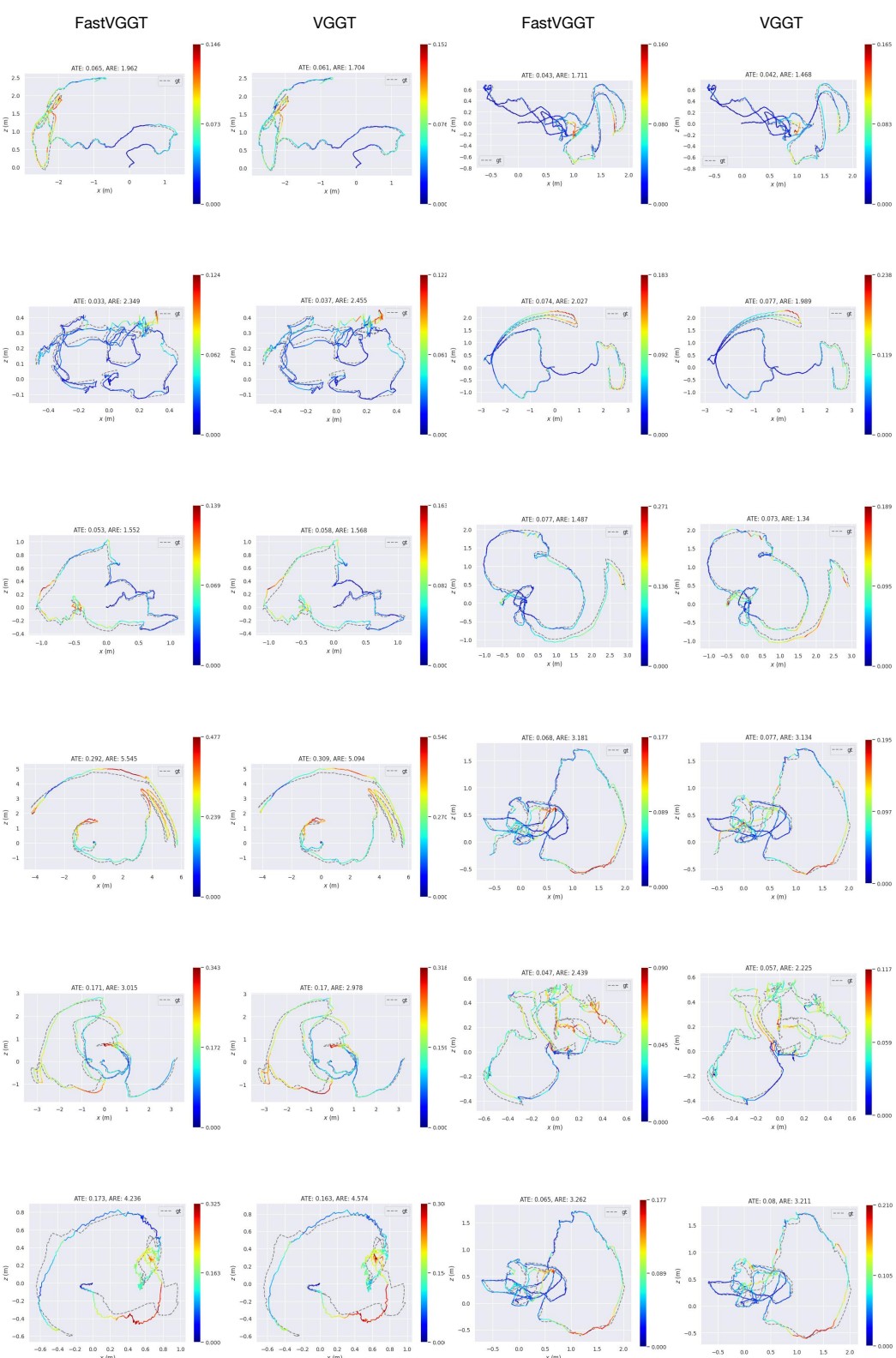

Figure 13: Additional visualizations of pose estimation results on the ScanNet dataset.

