# OpenReview forum: "FastVGGT: Fast Visual Geometry Transformer"
_ICLR.cc/2026/Conference — ICLR 2026 Poster_

### Official Review · Reviewer_rRZP · 2025-10-20

**Soundness:** 4
**Presentation:** 3
**Contribution:** 2
**Rating:** 6
**Confidence:** 4

**Summary:**

The paper concerns an efficient extension to VGGT, targeting the attention bottleneck in the original design. The authors perform a thorough analysis and find redundancy in the token representations. The authors call this "token collapse", where similarities for clusters of tokens end up becoming more or less indistinguishable, representing a clear redundancy. The authors exploit this by implementing structural merging via an explicit partitioning strategy. While the underlying ideas for reducing the complexity is not entirely new; the paper introduces specific mechanisms that seem well aligned with the objective. The authors highlight that their method achieves a $4\times$ speedup on 1000 image sequences, all while maintaining reconstruction quality.

**Strengths:**

1. The paper solves an important computational challenge with VGGT training and inference, which is of significant interest.
2. The methodology and experimental setup is well designed, clearly written, and comes across as technically competent engineering.
3. The technical contribution is intuitive, and well aligned with the goals the authors establish. Redefining the merging cycle for multi-view consistency, anchoring to the first frame and protecting salient tokens comes across as thoughtful and insightful modelling choices.

On the whole, the contributions are well motivated and the paper comes across as a well presented, with clear applications for a prominent contemporary modelling framework.

**Weaknesses:**

1. While the work solves an important problem with VGGT, the work is somewhat incremental; though in a technically competent and practically valuable sense. This is a minor, high level weakness, but has some implications, detailed below.
2. The proposed merging mechanism is presented as a novel architectural contribution, but has close parallels in existing literature. The authors mention the cosine-based assignment and replication in ToMeSD [1]; an extension of ToMe [2] for diffusion. However, other approaches such as TokenLearner [3], SPiT [4], and ATC [5] are other works that apply similar mechanisms for perceptual grouping and exploiting redundancy, explicitly setting out to capture partitions with constraints of connectivity in 2D images. More tangentially, the averaging and landmark selection procedure can be said to correspond to Nyström-like kernel compression and feature sketching [6; 7], and can also arguably related to general low-rank approximation. This reviewer emphasises that while the paper’s integration of these techniques into 3D feed-forward geometry transformers is valuable, the current phrasing makes the work come across as a somewhat *isolated discovery*. In this reviewers opinion, the contributions of the paper would be clearer if the authors explicitly positioned the contribution within the existing body of work on the topic.
3. Relatedly, an analysis on how the constrains used in the work (e.g. first-frame anchoring, salient protection) differ / synergise with existing works would benefit the field of token merging. Discussing how / why "geometry-aware" settings align / demand this specific adaptation would also be an interesting addition, and strengthen the work; particularly if the work is contrasted with existing methods.
4. The paper is a little sparse in discussing the inevitable failure cases and practical limitations, as well as intuition for hyperparameters (merging ratio). The effect of merging naturally induces a loss of information, and failure cases (if applicable) would not be unnatural, nor surprising given the improved efficiency of the approach.
5. Certain notation could be clarified slightly. L314 has examples of "src" and "dst" that is written out in plain text, which makes the notation hard to parse when placed in the prose. This is a minor weakness, but addressing this would improve the readability of the manuscript.

---

[1] [Token Merging for Fast Stable Diffusion, Bolya & Hoffman, 2023](https://arxiv.org/abs/2303.17604)

[2] [Token Merging: Your ViT But Faster, Bolya et al., 2022](https://arxiv.org/abs/2210.09461)

[3] [TokenLearner: Adaptive Space-Time Tokenization for Videos, Ryoo et al., 2021](https://openreview.net/forum?id=z-l1kpDXs88)

[4] [A Spitting Image: Modular Superpixel Tokenization in Vision Transformers, Aasan et al., 2024](https://arxiv.org/abs/2408.07680)

[5] [Agglomerative Token Clustering, Haurum et al., 2024](https://arxiv.org/abs/2409.11923)

[6] [Using the Nyström Method to Speed Up Kernel Machines, Williams & Seeger, 2001](https://papers.nips.cc/paper_files/paper/2000/hash/19de10adbaa1b2ee13f77f679fa1483a-Abstract.html)

[7] [Nyströmformer: A Nyström-Based Algorithm for Approximating Self-Attention, Xiong et al., 2021](https://arxiv.org/abs/2102.03902)

**Questions:**

1. The authors uses the term “geometry-aware” liberally, but the mechanism operates entirely in feature space. Anchoring to the first frame and preserving salient tokens is plausible, but not proven to maintain geometric invariants. This reviewer is left wondering what the authors mean when stating that the model is "geometry-aware", and whether the authors considers this worthwhile defining more clearer in the manuscript?
2. The method’s efficiency stems from token merging, yet comparisons are made only to 3D reconstruction baselines, not to efficient-transformer methods like ToMe, Nyströmformer, or even Performer [8]. The question is then; why not compare against generic attention approximations? Of course, this may not be trivial, but a discussion around why such a comparison is not included would be prudent.
3. The merging ratio is treated as a fixed hyperparameter with minimal analysis. Do the authors have any insights to share with practitioners who would like to tune this in more specific applications?
4. The paper reports mean metrics but is missing a failure analysis. Are errors gradual or catastrophic? Do high-parallax / low-texture regions induce failure compared to the original? Does merging induce smoothing, or loss of information, as one would expect?

While not as much a question, but this reviewer emphasises that the manuscript would be strengthened by extending the related work section to include previous works on partitioning for reducing token redundancy.

---

[8] [Rethinking Attention with Performers, Choromanski et al., 2020](https://arxiv.org/abs/2009.14794)

---

> ### Author Response · Authors · 2025-11-24
> **Response to rRZP (1/3)**
>
> We thank the reviewer for the careful review and the constructive discussions.
>
> **Q1: Clarify our contribution**&#x20;
>
> **A1:** We appreciate the comprehensive set of references related to model compression and acceleration. We have reviewed all these works and added the corresponding references and discussion in Section 2.2. In addition, following your suggestion, we have provided a clearer explanation of our contribution in Section 1. It is worth emphasizing that although 2D token merging also relies on similarity cues, token collapse in VGGT reveals much higher redundancy, allowing the model to safely adopt much more aggressive merging ratios and achieve greater speedups. For instance, ToMeSD degrades noticeably once the merging ratio exceeds 0.3, whereas FastVGGT preserves baseline-level performance even at a ratio of 0.9. This distinction is now clarified in Section 2.2.
>
> **Q2: Discussion on the Geometry-Aware Setting**&#x20;
>
> **A2:** The Geometry-Aware Setting comprises two components: first-frame anchoring and salient-token protection. As discussed in Section 4.5 VGGT’s global attention effectively performs high-level keypoint extraction and correspondence establishment, mirroring long-standing principles from SLAM and SfM. Importantly, the matching cues learned by VGGT are highly redundant. Our method leverages this insight: by suppressing the number of keypoints extracted and used for matching, we both accelerate inference and mitigate error accumulation.
>
> From this perspective, the two geometry-aware components can be understood as follows:
> 1.First-frame anchoring. Since VGGT uses the first frame to define the world coordinate system, the keypoints in the first frame act as global landmarks throughout cross-view matching. We therefore keep all tokens from the first frame to maintain stable reference geometry.
> 2.Salient-token protection. The goal of salient tokens is to ensure a spatially uniform distribution of keypoints across the scene, preventing excessive merging in certain areas that would otherwise leave insufficient information for reliable matching.
>
> In essence, VGGT is fundamentally a matching-based model; therefore, both the first-frame anchoring and the salient-token protection improve the stability and reliability of the matching process.

---

> > ### Author Response · Authors · 2025-11-24
> > **Response to rRZP (2/3)**
> >
> > **Q3: Discussion of failure cases**&#x20;
> >
> > **A3:** Thank you for the constructive suggestion. While our method demonstrates substantial benefits on long sequences, applying an aggressive merging ratio to short sequences results in a noticeable decline in performance. This behavior is expected: token collapse becomes prominent only when sufficient frames are available, whereas short sequences inherently contain less redundancy, making aggressive merging less appropriate.
> >
> > To study this phenomenon more thoroughly, Appendix C includes additional experiments on ScanNet-50 with input lengths of 50, 30, 10, and 5 frames. The results consistently show that aggressive ratios (e.g., 0.9) degrade performance on short sequences, whereas a moderate ratio such as 0.3 yields accuracy comparable to the baseline. These findings not only validate your concern but also provide a clearer picture of how merging ratio interacts with sequence length, further motivating why our method primarily targets long-sequence acceleration.
> >
> > **Table A: Quantitative results of short sequence performance on the ScanNet-50 dataset.**
> >
> > | Ratio        | Input | ATE ↓     | ARE ↓      | RPE-rot ↓  | RPE-trans ↓ | Time ↓     |
> > | ------------ | ----- | --------- | ---------- | ---------- | ----------- | ---------- |
> > | **0.9**      | 50    | 0.143     | 3.474      | 1.820      | 0.087       | **2.230s** |
> > | **0.9**      | 30    | 0.172     | 3.750      | 2.826      | 0.150       | **1.348s** |
> > | **0.9**      | 10    | 0.250     | 5.316      | 6.225      | 0.369       | **0.628s** |
> > | **0.9**      | 5     | 0.452     | 18.626     | 22.989     | 0.682       | **0.433s** |
> > | **0.6**      | 50    | 0.142     | **3.447**  | 1.823      | 0.089       | 2.315s     |
> > | **0.6**      | 30    | 0.166     | **3.679**  | 2.722      | 0.146       | 1.373s     |
> > | **0.6**      | 10    | 0.246     | 4.460      | 5.148      | 0.306       | 0.637s     |
> > | **0.6**      | 5     | 0.452     | 17.905     | 22.179     | 0.673       | 0.440s     |
> > | **0.3**      | 50    | **0.137** | 3.466      | 1.698      | **0.079**   | 2.859s     |
> > | **0.3**      | 30    | 0.158     | 3.693      | 2.610      | 0.130       | 1.567s     |
> > | **0.3**      | 10    | 0.201     | 4.326      | 5.003      | 0.255       | 0.651s     |
> > | **0.3**      | 5     | **0.330** | 10.568     | 12.687     | **0.434**   | 0.443s     |
> > | **Baseline** | 50    | 0.138     | 3.512      | **1.685**  | **0.079**   | 3.478s     |
> > | **Baseline** | 30    | **0.154** | 3.705      | **2.543**  | **0.125**   | 1.977s     |
> > | **Baseline** | 10    | **0.197** | **4.131**  | **4.683**  | **0.247**   | 0.712s     |
> > | **Baseline** | 5     | 0.345     | **10.463** | **12.629** | 0.450       | 0.507s     |
> >
> > **Q4: Merging Ratio Selection**&#x20;
> >
> > **A4:** In developing our method, we explored various merging-ratio strategies, including adaptive ones. However, since our primary goal is long-sequence acceleration, we found that a fixed ratio of 0.9 already achieves strong performance in long-sequence settings. To keep the method simple and efficient, we therefore adopt 0.9 as the fixed merging ratio.
> >
> > **Q5: Clarify notation**&#x20;
> >
> > **A5:** We appreciate the reminder, and we have clarified the notation in Section 3.3.

---

> ### Author Response · Authors · 2025-11-24
> **Response to rRZP (3/3)**
>
> **Q6: Comparison with Efficient-Transformer Methods**
>
> **A6:** As you suggested, we have added experiments in Section 3.1 comparing our method with existing Efficient-Transformer approaches. Moreover, we want to view this comparison as preliminary evidence rather than a competing benchmark, because these methods were originally designed for 2D tasks. When directly applied to VGGT, they exhibit a substantial performance gap relative to the original VGGT baseline. This reinforces that naively adopting existing Efficient-Transformer designs is non-trivial, and that a merging strategy tailored specifically to VGGT is necessary.
>
> Furthermore, we carefully reviewed the excellent methods suggested by the reviewer. However, we aim to keep our approach entirely training-free to ensure practical applicability. Among the methods mentioned, only ATC\[1] operates in a training-free manner. Despite its strong performance on single-image 2D tasks, ATC is unsuitable for long-sequence acceleration: it relies on a top-down hierarchical procedure that computes a full similarity matrix, resulting in approximately 10^8 operations for just 100 frames—computationally prohibitive for our setting.
>
> Instead, we explored other methods with a similar spirit, PiToMe\[2], which selects higher-quality destination tokens. Building on PiToMe, our experiments further demonstrate that directly applying generic token-merging strategies remains non-trivial, underscoring the need for a merging design specifically adapted to VGGT’s architecture and long-sequence behavior.
>
> Finally, as requested, we have **expanded the related-work discussion** in the revised Section 2.2 to explicitly include the methods you mentioned as well as the additional approaches we newly surveyed.
>
> **Table E: Results of applying existing merging strategies. ToMe-S denotes the fixed-stride version of ToMe, and ToMe-R denotes the random-sampling version.**
>
> | Method | Ratio | CD↓ (300) | Time↓ (300) | CD↓ (100) | Time↓ (100) |
> | ------ | ----- | --------- | ----------- | --------- | ----------- |
> | ToMe-R | 0.3   | 0.451     | 102.8s      | 0.459     | 7.9s        |
> | ToMe-S | 0.3   | 0.633     | 98.4s       | 0.548     | 7.9s        |
> | PiToMe | 0.3   | 0.454     | 112.9s      | 0.450     | 8.7s        |
> | ToMe-R | 0.6   | 0.539     | 51.1s       | 0.485     | 6.5s        |
> | ToMe-S | 0.6   | 0.563     | 49.3s       | 0.523     | 6.4s        |
> | PiToMe | 0.6   | 0.489     | 61.6s       | 0.467     | 7.7s        |
> | ToMe-R | 0.9   | 0.582     | 22.8s       | 0.494     | 5.0s        |
> | ToMe-S | 0.9   | 0.708     | **21.2s**   | 0.628     | **4.8s**    |
> | PiToMe | 0.9   | 0.567     | 37.5s       | 0.522     | 7.2s        |
> | VGGT   | -     | **0.416** | 131.4s      | **0.423** | 9.1s        |
>
>
> **Summary**
>
> We hope our response can address all of your concerns raised in the rebuttal. If the reviewer still has any remaining questions or concerns, we would be glad to hear that.
>
> **Reference**
>
> \[1] Agglomerative Token Clustering. In ECCV 2024.
>
> \[2] Accelerating Transformers with Spectrum-Preserving Token Merging. In NeuIPS 2024.

---

> ### Comment · Reviewer_rRZP · 2025-11-27
>
> Thank you for the detailed response. This reviewer acknowledges the authors addresses most of the raised concerns of the review, and remain positive to the quality of the paper.
>
> Edit: The reviewer initially had some local issue with viewing the revised manuscript, but confirms that the authors have revised the manuscript with changes from the rebuttal. We thank the authors again and increase our score to reflect the changes.

---

> > ### Author Response · Authors · 2025-11-28
> >
> > We sincerely appreciate your thoughtful reading and recognition of our work.

---

### Official Review · Reviewer_f7WY · 2025-10-29

**Soundness:** 2
**Presentation:** 3
**Contribution:** 2
**Rating:** 4
**Confidence:** 4

**Summary:**

This paper proposes FastVGGT, a method to speed up Visual Geometry Transformers (VGGT) when handling long image sequences.
The authors find that the Global Attention layer causes heavy computation. To solve this, they introduce a training-free token pruning framework that keeps only the most useful tokens.
FastVGGT uses three steps: keeping the first-frame tokens for stability, preserving important tokens for details, and using region-based random sampling for balanced coverage.

Experiments show that FastVGGT achieves up to 4× faster inference and reduces error accumulation, making it more efficient and robust for long-sequence 3D vision tasks.

**Strengths:**

1. The paper tackles an important and practical problem — improving the inference efficiency of VGGT on long image sequences.

2. The proposed training-free token pruning strategy is simple yet effective. It significantly reduces computational cost while maintaining nearly the same accuracy.

3. The identification of the “token collapse” phenomenon in VGGT’s Global Attention is insightful.

4. The paper is clearly written and well-structured, with intuitive motivation and clear methodology.

**Weaknesses:**

1. The improvement is very limited. As shown in Tables 2, 3, and 4, FastVGGT shows almost no accuracy gain compared to VGGT*, and in some cases even performs slightly worse. This weak improvement reduces the overall significance of the contribution.

2. A straightforward expectation is that FastVGGT should reduce memory usage during long-sequence inference by pruning redundant tokens. However, the paper does not provide any experimental evidence or quantitative analysis on memory consumption. As a result, it remains unclear whether FastVGGT actually achieves significant memory savings when performing global attention over long sequences.

3. Comparisons are limited. Most baselines are just VGGT variants. In Table 4 the authors report Pi3 as OOM; it would be more informative to implement an optimized variant of Pi3 (analogous to VGGT*) so a direct, fair comparison is possible.

**Questions:**

1. In Tables 2, 4, and 5, the meaning of the bold numbers is unclear. If bold font is intended to indicate the best performance, there are several cases where FastVGGT performs worse than VGGT* but is still highlighted. This inconsistency creates confusion and raises concerns about the fairness and clarity of result presentation.

2. The claimed advantage in handling long sequences is not clearly demonstrated. For example, in Table 2, the reconstruction quality with 1000 frames is even worse than with 500 frames, which contradicts the paper’s stated goal of improving long-sequence performance. This raises doubts about whether FastVGGT truly provides benefits for very long input sequences.

**Details Of Ethics Concerns:**

None.

---

> ### Author Response · Authors · 2025-11-24
> **Response to f7WY (1/2)**
>
> We thank the reviewer for the careful review and the constructive discussions.
>
> **Q1: Limited improvement**
>
> **A1:** Thank you for pointing this out. The primary improvement of our work lies in significantly accelerating long-sequence processing. From the perspective of performance rather than efficiency, the gains on short sequences are naturally limited: token-collapse redundancy does not accumulate enough to introduce meaningful noise into VGGT, so accuracy differences remain small. This likely explains the modest improvements observed on 100-frame sequences.
>
> When moving to long sequences, typically above 500 frames, the situation changes. FastVGGT maintains stable reconstruction accuracy, while VGGT begins to degrade in both accuracy and feasibility due to memory constraints. In these long-range scenarios, FastVGGT achieves:
>
> 1.  Strong efficiency improvements in both computation time and memory
> 2.  Comparable or clearly better accuracy than VGGT
> 3.  Providing 2–4× speed-up in both latency and memory consumption
>
> Therefore, the significance of our contribution lies in enabling practical long-sequence processing without sacrificing accuracy. We have revised the paper to emphasize this motivation.
>
>
> **Q2: Explanation of memory usage**
>
> **A2:** As recommended, we added Figure 7 in the Section 3.4 to show the quantitative memory-consumption comparison between VGGT\* and the original VGGT. Notably, the reduction in memory usage is not a result of token merging; instead, it stems from our observations and optimizations of the VGGT's framework (see Section 3.4), including the removal of redundant intermediate results. This is an insightful question about why token merging does not directly lead to memory savings. In principle, token merging reduces memory usage by lowering the number of tokens involved in attention computation. However, we observe no meaningful token similarity within the frame-attention layers. Furthermore, since VGGT alternates global-attention and frame-attention blocks, merged tokens must be restored to full resolution before entering each frame-attention layer, resulting in no reduction in token count. We appreciate the suggestion and have provided a clearer explanation in Section 3.4.
>
> **Table F: Comparison of GPU memory between the original VGGT and VGGT**\*
>
> | Inputs Frames | VGGT\*   | VGGT     |
> | ------------- | -------- | -------- |
> | 300           | 15.10 GB | 78.39 GB |
> | 250           | 12.94 GB | 68.91 GB |
> | 200           | 10.79 GB | 56.34 GB |
> | 150           | 8.63 GB  | 43.78 GB |
> | 100           | 6.48 GB  | 31.59 GB |
> | 50            | 4.35 GB  | 18.54 GB |
>
> **Q3: Memory optimization for Pi3**
>
> **A3:** Thank you for the helpful suggestion. We have carefully examined Pi3 as a potential baseline. Although Pi3 and VGGT share the same 18-block attention backbone, the two systems differ fundamentally in their inference design. Pi3 retains only the final-layer outputs of each block, without preserving intermediate activations for later geometry reasoning. As a result, Pi3 already avoids the major source of memory redundancy that FastVGGT targets in VGGT.

---

> > ### Author Response · Authors · 2025-11-24
> > **Response to f7WY (2/2)**
> >
> > **Q4: Result presentation**
> >
> > **A4:** We regret the ambiguity introduced by the earlier presentation of our results. We have updated all tables to improve clarity with the best results highlighted in bold and the second-best results underlined. We appreciate the reviewer’s constructive suggestion.
> >
> > **Q5: Significance of supporting long-sequence**
> >
> > **A5:** Thank you for the insightful comment. We fully agree that longer inputs should ideally provide richer geometric cues and ultimately yield higher reconstruction quality. However, current VGGT-style architectures struggle when the sequence becomes very long: redundant attention and accumulated noise can overshadow the benefit of seeing more frames. This architectural limitation explains why the accuracy using 1000 frames does not always surpass that of 500 frames.
> >
> > Importantly, this degradation is not introduced by our acceleration strategy. Instead, FastVGGT makes it possible to run these ultra-long sequences efficiently and stably, which in turn reveals this inherent weakness of existing architectures — a weakness that was previously hidden because baseline VGGT often cannot complete inference at such lengths due to memory constraints and error accumulation.
> >
> > Compared with VGGT under the same settings, FastVGGT:
> >
> > 1.  prevents more severe degradation seen in VGGT under the same setting
> > 2.  extends the feasible input range far beyond that of VGGT
> > 3.  partially alleviates collapse-induced error propagation
> >
> > Thus, our contribution is two-fold:
> >
> > 1.  practically enabling long-sequence inference today, and
> > 2.  surfacing a clear research direction for fully harnessing long-range information in future architectures.
> >
> > While there is still a long way to go before extremely long sequences can consistently outperform shorter ones, FastVGGT provides both the capability and the conceptual foundation needed to make such progress attainable. We have added this clarification in Section 5.
> >
> > **Summary**
> > We hope our response can address all of your concerns raised in the rebuttal. If the reviewer still has any remaining questions or concerns, we would be glad to hear that.

---

> ### Author Response · Authors · 2025-11-27
> **Thanks for your recognition of the value of this work, we sincerely look forward to your feedback.**
>
> Dear Reviewer f7WY,
>
> Thank you once again for your valuable comments on our submission. As the discussion phase is approaching its end, we would like to kindly confirm whether we have sufficiently addressed all of your concerns (or at least part of them). Should there be any remaining questions or areas requiring further clarification, please do not hesitate to let us know. If you are satisfied with our responses, we would greatly appreciate your consideration in adjusting the evaluation scores accordingly.
>
> We sincerely look forward to your feedback.

---

### Official Review · Reviewer_aGnr · 2025-10-30

**Soundness:** 2
**Presentation:** 1
**Contribution:** 2
**Rating:** 2
**Confidence:** 4

**Summary:**

FastVGGT accelerates VGGT's 3D reconstruction inference by applying training-free token merging to the Global Attention bottleneck. Through strategic token partitioning—preserving reference frames, retaining salient tokens, and uniform sampling—it achieves 4× speedup on 1000-image inputs while maintaining reconstruction quality and mitigating error accumulation in long sequences.

**Strengths:**

1. The method achieves substantial 4× speedup on 1000-image inputs while maintaining or even improving reconstruction quality, and effectively mitigates error accumulation in long-sequence scenarios. This addresses a critical bottleneck in feed-forward 3D reconstruction models and demonstrates clear real-world applicability.

2. The approach requires no model retraining or fine-tuning, making it immediately deployable to existing VGGT systems. The comprehensive experiments across multiple benchmarks (ScanNet, 7 Scenes, NRGBD) validate its effectiveness and generalizability across diverse 3D reconstruction tasks.

**Weaknesses:**

1. This paper suffers from insufficient experimental details and lacks fair comparisons. Critical hyperparameters are inadequately documented: while the ablation study (Table 7) examines merging ratios of 0.3, 0.6, and 0.9, the main experiments (Tables 2-5) do not explicitly state which configuration was used.

2. More problematically, Table 1 presents baseline comparisons with naive merging strategies (random and fixed-stride sampling) but provides inconsistent experimental conditions. The random sampling uses merging ratios (r=0.5, 0.8) while fixed-stride sampling uses stride values (s=5, 8), these setting do not align with any setting of the proposed method. Furthermore, what is the testing data? The resolution? The number of input frames? The paper provides no explanation for why certain configurations were chosen or how they relate to the proposed method's settings. This makes it nearly impossible to assess whether the improvements stem from the proposed partitioning strategy or simply from different hyperparameter choices. A fair comparison would require evaluating all methods under equivalent compression rates with clearly documented settings.

3. The technical contribution of this paper is limited. The core proposal consists of three token partitioning strategies: (1) designating first-frame tokens as destinations, (2) retaining high-norm tokens as salient, and (3) applying region-based random sampling. While these choices are reasonable for 3D reconstruction tasks, they represent relatively straightforward engineering adaptations rather than fundamental algorithmic innovations. The token merging mechanism itself is directly borrowed from existing 2D vision methods (ToMe, ToMeSD) without architectural modifications. The main contribution appears to be identifying where to apply existing techniques rather than developing novel methods.

**Questions:**

See Weaknesses.

---

> ### Author Response · Authors · 2025-11-24
> **Response to aGnr (1/2)**
>
> We thank the reviewer for the careful reading of our paper and the detailed constructive comments.
>
> **Q1: The merging ratio was not clearly stated.**
>
> **A1:** We have updated Section 4.1 to include the merging ratio used in our experiments. All main results (Tables 2–7) employ a merging ratio of 0.9. We apologize that in the previous version this information appeared only in the ablation study, making it easy to miss and lacking an explicit statement. We thank the reviewer for the careful feedback, which helped improve the clarity and reproducibility of our paper.
>
> **Q2: Missing fair comparison with naive merging strategies.**
>
> **A2:** We have updated Table 1 to ensure consistency in the experimental conditions. On the ScanNet-50 dataset, we compare the baseline performance with different merging ratios (0.9, 0.6, 0.3) and different sequence lengths (300, 100) across random-based ToMe, fixed-stride ToMe, and the more advanced PiToMe method. The results clearly demonstrate that directly applying existing merging strategies is non-trivial, as they exhibit substantial performance gaps compared to the baseline. This underscores the necessity of developing a merging strategy tailored to the VGGT architecture.
>
> The previous version of Table 1 contained preliminary exploratory experiments, which reported results from ten scenes of ScanNet-50 using 200-frame inputs. We sincerely regret that these preliminary settings were incomplete and not fully aligned with the main experimental configuration. Thank you for identifying this inconsistency—your feedback has greatly strengthened the rigor of our work.
>
> **Table E: Results of applying existing merging strategies. ToMe-S denotes the fixed-stride version of ToMe, and ToMe-R denotes the random-sampling version.**
>
> | Method | Ratio | CD↓ (300) | Time↓ (300) | CD↓ (100) | Time↓ (100) |
> | ------ | ----- | --------- | ----------- | --------- | ----------- |
> | ToMe-R | 0.3   | 0.451     | 102.8s      | 0.459     | 7.9s        |
> | ToMe-S | 0.3   | 0.633     | 98.4s       | 0.548     | 7.9s        |
> | PiToMe | 0.3   | 0.454     | 112.9s      | 0.450     | 8.7s        |
> | ToMe-R | 0.6   | 0.539     | 51.1s       | 0.485     | 6.5s        |
> | ToMe-S | 0.6   | 0.563     | 49.3s       | 0.523     | 6.4s        |
> | PiToMe | 0.6   | 0.489     | 61.6s       | 0.467     | 7.7s        |
> | ToMe-R | 0.9   | 0.582     | 22.8s       | 0.494     | 5.0s        |
> | ToMe-S | 0.9   | 0.708     | **21.2s**   | 0.628     | **4.8s**    |
> | PiToMe | 0.9   | 0.567     | 37.5s       | 0.522     | 7.2s        |
> | VGGT   | -     | **0.416** | 131.4s      | **0.423** | 9.1s        |

---

> > ### Author Response · Authors · 2025-11-24
> > **Response to aGnr (2/2)**
> >
> > **Q3: Lack of detailed experimental configuration.**
> >
> > **A3:** We have reviewed all experiments and added the complete experimental settings in Section 4.1, including the testing data, input resolution, number of input frames, and merging-ratio configurations. These additions ensure that all experiments are fully reproducible and that the evaluation conditions across different tables and figures are transparent and consistent. We sincerely appreciate your constructive suggestion.
> >
> > **Q4: Limited technical contribution**
> >
> > **A4:** We thank the reviewer for the detailed comments. We would like to clarify that our main contribution is uncovering token collapse in VGGT and introducing a specialized merging strategy for acceleration. Prior VGGT works focus mainly on architectural scaling or accuracy improvement, but none have investigated why performance degrades when the sequence becomes long, nor what internal mechanism restricts scalability.
> >
> > Our analysis reveals that redundant, low-saliency background tokens dominate the global-attention aggregation, gradually suppressing structurally informative tokens and ultimately degrading correspondence quality. This “token-collapse” phenomenon is visualized in Section 3.1 and quantitatively analyzed in Section 4.5, and, to the best of our knowledge, has not been previously discussed.
> >
> > Grounded in this insight, we introduce a targeted and interpretable merging strategy consisting of:
> > 1\)First-frame anchoring to maintain a consistent global geometry reference;
> > 2\)Salient token preservation to retain reliable correspondence cues;
> > 3\)Region-aware sampling to ensure geometric coverage in large smooth areas.
> >
> > While each component is intentionally lightweight, their synergy directly exploits the collapse phenomenon, enabling VGGT to process much longer sequences with improved performance and reduced computation. These choices are not arbitrary engineering heuristics; rather, they are explicitly derived from our observations and subsequent analysis, and the ablation studies clearly demonstrate their necessity.
> >
> > Importantly, research significance in deep learning is often driven by revealing mechanisms rather than complexity of modules. For example, ResNet\[1] introduced a structurally simple identity skip connection, yet its impact lies in uncovering the cause of degradation in deep networks and transforming modern architectures. Likewise, our work improves the understanding of why token interactions break down in long-range matching, and provides a simple yet principled remedy.
> >
> > **Summary**
> > We have revised the paper to strengthen this motivation, add comparisons to alternative merging schemes, and expand related discussion accordingly. We appreciate the reviewer’s feedback, which helped us better articulate the conceptual and technical value of our design.
> >
> > **Reference**
> >
> > \[1] Deep residual learning for image recognition. In CVPR 2016.

---

> ### Author Response · Authors · 2025-11-27
> **Thanks for your recognition of the value of this work, we sincerely look forward to your feedback.**
>
> Dear Reviewer aGnr,
>
> Thank you once again for your valuable comments on our submission. As the discussion phase is approaching its end, we would like to kindly confirm whether we have sufficiently addressed all of your concerns (or at least part of them). Should there be any remaining questions or areas requiring further clarification, please do not hesitate to let us know. If you are satisfied with our responses, we would greatly appreciate your consideration in adjusting the evaluation scores accordingly.
>
> We sincerely look forward to your feedback.

---

> ### Comment · Reviewer_aGnr · 2025-11-28
>
> Dear authors,
>
> Thank you for your detailed reply. I think the response has addreesed my concern about the experiment settings and made the contribution more significant. I would raise the score if possible.

---

### Official Review · Reviewer_SPNW · 2025-10-31

**Soundness:** 3
**Presentation:** 3
**Contribution:** 3
**Rating:** 4
**Confidence:** 4

**Summary:**

This paper aims to solve the computational bottleneck of the SOTA model VGGT when processing long image sequences. The authors identify the bottleneck as computational redundancy in the Global Attention layer, a phenomenon they term "token collapse." To address this, the paper proposes FastVGGT, a training-free token merging framework. Its core is a token partitioning strategy specifically tailored for 3D geometry tasks (preserving the reference frame, retaining salient tokens, and uniform sampling). Experiments show that this method achieves a 4x speedup when processing 1000-frame sequences while simultaneously improving reconstruction accuracy and reducing error accumulation.

**Strengths:**

The practical value of this paper is extremely high. It proposes a training-free, plug-and-play method that solves a critical performance bottleneck in the SOTA model VGGT. Achieving up to 4x inference speedup makes large-scale 3D reconstruction more feasible for practical applications.

Most impressively, the method not only accelerates inference but also significantly improves accuracy on long sequences. By mitigating error accumulation, it outperforms the baseline VGGT on both pose estimation and 3D reconstruction tasks, which is a very strong and non-trivial contribution.

The method demonstrates good originality. Its three-part token partitioning strategy (preserving the reference frame, retaining salient tokens, uniform sampling) is specifically designed for the characteristics of 3D geometry tasks (like cross-frame correspondence and a global coordinate system), rather than just being a simple application of standard 2D token merging.

**Weaknesses:**

W1: Insufficient discussion of the performance trade-off on short sequences. The paper's strategy and hyperparameters (e.g., 90% merge ratio) are clearly optimized for long sequences (e.g., 500-1000 frames). However, on short sequences (100 frames), the performance is mixed, even performing slightly worse than the baseline on some key metrics (e.g., RPE-rot and RPE-trans in Table 5). The authors acknowledge this in Appendix E, but a deeper analysis is missing from the main experimental section.

W2: Disconnect between motivation and implementation for "salient token" selection. The motivation for "salient tokens" is described as preserving unique, "keypoint-like" features, and the paper mentions that a "top-k based on token norms" strategy was considered. However, a simple "fixed-stride sampling" was ultimately used for efficiency. The claim that both are "comparably accurate" is unsubstantiated (no data in ablations), which creates a disconnect between the final design and its original motivation (to preserve the most unique features).

W3: Robustness to "reference frame" quality is questionable. The entire method relies heavily on the first frame as the "world coordinate system" and global anchor, designating all its tokens as high-priority dst tokens. The paper does not discuss at all how the model would perform if this first frame is of poor quality (e.g., blurry, occluded, or unrepresentative of the scene). This reliance on a single, arbitrarily chosen frame could be a robustness liability.

W4: Evidence for "token collapse" is primarily qualitative. The paper's core motivation—the "token collapse" phenomenon—is primarily argued via visualizations of attention maps in Figure 3. While these maps look similar, this is only a qualitative observation. The paper lacks a more rigorous quantitative analysis (e.g., calculating average cosine similarity or feature variance between tokens) to prove the degree of redundancy, which weakens the force of its core motivation.

W5: Justification for the fixed-stride sampling strategy. Using "fixed-stride sampling" to select "salient tokens" seems like an arbitrary choice. "Keypoints" in a scene are unlikely to be uniformly distributed on an image grid. This sampling method runs counter to the motivation of preserving the "most unique" features and is potentially less effective than the (W2-mentioned) top-k strategy.

**Questions:**

Q1 : You determined an aggressive 90% merge ratio starting from Block 0, based on 500-frame sequences (Table 7). Is this fixed hyperparameter combination optimal for all tested sequence lengths (100, 300, 1000 frames)? Could the slight performance drop on 100-frame sequences (Table 5 RPE) be resolved by using a more conservative (lower) merge ratio? Does this suggest that an adaptive merge strategy (e.g., merge ratio $r$ as a function of sequence length $N$) would be a better choice?

Q2 : Could you please provide a specific quantitative comparison of accuracy (CD/ATE) and inference time between (a) the top-k based on token norms strategy and (b) the final fixed-stride sampling strategy for "salient token" selection? (You claimed they were "comparably accurate" in Section 3.2). This is crucial for justifying your final design choice.

Q3 : Your method is highly dependent on using the first frame as a global reference. Did you test the method's robustness to the quality of this first frame? For example, how much does FastVGGT's performance degrade if the first frame is blurry or heavily occluded? Have you considered alternative reference selection strategies, such as choosing the sharpest frame in the sequence (via some quality metric) as the reference, rather than blindly using the first one?

Q4 : For your "token collapse" motivation, did you perform any quantitative measurements beyond the qualitative visualizations in Figure 3 to confirm this phenomenon? For example, by calculating the average cosine similarity between different token feature vectors in the Global Attention layers and showing that it increases in deeper layers? This would provide stronger support for your motivation.

Q5 : The most exciting result in the paper is the improvement in accuracy on long sequences, which you attribute to "mitigating error accumulation." Could you please explain mechanistically, in more detail, why removing tokens (even redundant ones) helps prevent error propagation? My hypothesis is: by merging redundant information in early layers (starting from Block 0), the model is forced to form a more compact and robust global representation, which prevents "over-thinking" on noise and redundant features in later layers, ultimately reducing drift. Do you agree with this hypothesis, or do you have other insights?

---

> ### Author Response · Authors · 2025-11-24
> **Response to SPNW (1/3)**
>
> We thank the reviewer for the thorough review and the insightful feedback and discussions.
>
> **Q1: Merging ratio analysis for short sequences**
>
> **A1:** **(1) Hyperparameter settings:** All experiments in the main paper (Tables 2–6) use a fixed merging ratio of **0.9**, and we have added this clarification in Section 4.1. We did examine more adaptive ratio strategies during the early development stage, but our experiments indicated that for the **long-sequence settings** that motivate this work (typically >100 frames), a fixed and relatively aggressive ratio already provides stable and effective performance.
>
> Because the goal of this paper is to **highlight the token-collapse phenomenon in VGGT** and to propose a **simple, training-free, and efficient** mitigation strategy, we adopt the fixed ratio of 0.9 throughout the main experiments.
>
> **(2) Experiments on short sequences:** We have added experiments in Appendix C evaluating different merging ratios on ScanNet-50 with input lengths of 50, 30, 10, and 5 frames. The results show that aggressive ratios (e.g., 0.9) may lead to performance degradation on very short sequences. This behavior is expected: the correspondence redundancy induced by token collapse is less pronounced in short sequences, making an aggressive merging ratio more likely to discard valuable information. Consequently, the benefit of merging becomes minimal when only a few frames are available.  In such settings, a milder ratio such as 0.3 preserves baseline-level accuracy.
>
> We would like to emphasize that the **primary motivation of this work is to accelerate long-sequence processing**, where token collapse reflects a significant redundancy and where a higher merging ratio provides clear benefits in both efficiency and stability. Although FastVGGT can also speed up short sequences, the potential improvements are naturally limited because short-range VGGT already runs efficiently.
>
> **Table A: Quantitative results of short sequence performance on the ScanNet-50 dataset.**
>
> | Input  | Ratio    | ATE ↓     | ARE ↓      | RPE-rot ↓  | RPE-trans ↓ | Time ↓     |
> | ------ | -------- | --------- | ---------- | ---------- | ----------- | ---------- |
> | **50** | 0.9      | 0.143     | 3.474      | 1.820      | *0.087*     | **2.230s** |
> |        | 0.6      | 0.142     | **3.447**  | 1.823      | 0.089       | *2.315s*   |
> |        | 0.3      | **0.137** | *3.466*    | *1.698*    | **0.079**   | 2.859s     |
> |        | Baseline | *0.138*   | 3.512      | **1.685**  | **0.079**   | 3.478s     |
> | **30** | 0.9      | 0.172     | 3.750      | 2.826      | 0.150       | **1.348s** |
> |        | 0.6      | 0.166     | **3.679**  | 2.722      | 0.146       | *1.373s*   |
> |        | 0.3      | *0.158*   | *3.693*    | *2.610*    | *0.130*     | 1.567s     |
> |        | Baseline | **0.154** | 3.705      | **2.543**  | **0.125**   | 1.977s     |
> | **10** | 0.9      | 0.250     | 5.316      | 6.225      | 0.369       | **0.628s** |
> |        | 0.6      | 0.246     | 4.460      | 5.148      | 0.306       | *0.637s*   |
> |        | 0.3      | *0.201*   | *4.326*    | *5.003*    | *0.255*     | 0.651s     |
> |        | Baseline | **0.197** | **4.131**  | **4.683**  | **0.247**   | 0.712s     |
> | **5**  | 0.9      | 0.452     | 18.626     | 22.989     | 0.682       | **0.433s** |
> |        | 0.6      | 0.452     | 17.905     | 22.179     | 0.673       | *0.440s*   |
> |        | 0.3      | **0.330** | *10.568*   | *12.687*   | **0.434**   | 0.443s     |
> |        | Baseline | *0.345*   | **10.463** | **12.629** | *0.450*     | 0.507s     |

---

> > ### Author Response · Authors · 2025-11-24
> > **Response to SPNW (2/3)**
> >
> > **Q2: Quantitative comparison of salient token selection schemes**
> >
> > **A2:** As suggested, we have added in Section 4.4 a comparison between the top-k–based and fixed-stride sampling strategies for selecting salient tokens. On ScanNet-50, we evaluate performance and efficiency and further analyze the effect of selecting different numbers of salient tokens. The results show that the fixed-stride strategy noticeable improves efficiency while maintaining comparable performance. Considering both accuracy and efficiency, we ultimately adopt the fixed-stride (10%) strategy.
> >
> > To further analyze the effects of the two strategies, we added additional experiments in Appendix E.2 to visualize the spatial distribution of salient tokens. We observe that Top-K performs well in shallow blocks by accurately capturing semantically important regions of the scene, but as the block depth increases, the selected salient tokens become increasingly concentrated. In contrast, although the fixed-stride strategy is less precise in identifying key token locations, it consistently distributes salient tokens uniformly across different blocks.
> >
> > These experiments bridge the missing justification in our original submission and strengthen the design choice. We appreciate the reviewer’s feedback, which has helped us improve the coherence and rigor of the paper.
> >
> > **Table B: Quantitative results of salient token selection strategies on the ScanNet-50 dataset.**
> >
> > | Method     | CD↓ (500) | Time↓ (500) | CD↓ (300) | Time↓ (300) | CD↓ (100) | Time↓ (100) |
> > | ---------- | --------- | ----------- | --------- | ----------- | --------- | ----------- |
> > | TopK-15%   | 0.421     | 80.3s       | **0.410** | 43.8s       | 0.432     | 7.5s        |
> > | TopK-10%   | 0.423     | 73.8s       | 0.423     | 35.3s       | 0.428     | 6.7s        |
> > | TopK-5%    | 0.438     | 68.1s       | 0.412     | 31.4s       | 0.445     | 6.1s        |
> > | Stride-15% | 0.430     | 62.8s       | 0.429     | 28.5s       | 0.434     | 6.0s        |
> > | Stride-10% | **0.411** | 55.2s       | 0.416     | 23.8s       | **0.426** | 5.4s        |
> > | Stride-5%  | 0.425     | **52.5s**   | 0.427     | **22.4s**   | 0.438     | **4.9s**    |
> >
> > **Q3: Robustness to the first-frame input**
> >
> > **A3:** In Appendix D, we include additional experiments that evaluate robustness to first-frame blur and occlusion, as well as more extensive visualizations. On ScanNet-50, we use a blur kernel of size 201 (σ = 30) and apply a 40×40 mask that occludes 50% or 80% of the first frame. Interestingly, FastVGGT achieves better performance than the baseline VGGT even in these challenging scenarios. To further understand this behavior, we quantify the contribution of first-frame dst tokens: with 500 input frames, only 1.96% of merged dst tokens originate from the first frame; with 100 input frames, the proportion is 9.17%. These statistics suggest that, if the first frame has poor quality, token merging will predominantly rely on matching confidence from later frames, which may help mitigate the negative impact of early-frame degradation. We have incorporated these results and analysis to better support the robustness claim.
> >
> > **Table C: Quantitative results of first-frame occlusion or blur on the ScanNet-50 dataset.**
> >
> > | First Frame   | Method   | 500       | 300       | 100       |
> > | ------------- | -------- | --------- | --------- | --------- |
> > | Occlusion-50% | VGGT     | 0.501     | 0.482     | 0.475     |
> > |               | FastVGGT | **0.471** | **0.452** | **0.441** |
> > | Occlusion-80% | VGGT     | 0.514     | 0.503     | 0.488     |
> > |               | FastVGGT | **0.476** | **0.468** | **0.459** |
> > | Blur          | VGGT     | 0.509     | 0.495     | 0.470     |
> > |               | FastVGGT | **0.477** | **0.461** | **0.454** |

---

> ### Author Response · Authors · 2025-11-24
> **Response to SPNW (3/3)**
>
> **Q4: Quantitative analysis of token collapse**
>
> **A4:** Following the reviewer’s suggestion, Section 4.5 B now includes additional experiments that use average cosine similarity to analyze VGGT’s token-collapse phenomenon. The experiments are performed on ScanNet-50, adopting the same setup as in Figure 3. Specifically, we compute the similarity metric as the average pairwise cosine similarity across the six tokens (0, 500, 1041, 1541, 10410, 10910). Cosine similarity ranges from −1 to 1, where 1 means highly similar token-attention patterns, 0 means uncorrelated, and −1 means opposite. The experiments show that the global attention exhibits substantial similarity across most blocks, but a noticeable drop occurs around Blocks 1 and 14.
>
> Interestingly, even in layers where the token similarity is relatively low, applying an aggressive merging ratio (0.9) does not harm the final performance. To further analyze this phenomenon, we refer to conclusions drawn from our attention-map visualizations: VGGT’s global attention is fundamentally driven by feature matching, with high activation values concentrated around keypoint-like regions. Therefore, even though the token similarity in the middle blocks is not particularly high, we still apply an aggressive merging ratio in these layers, effectively forcing the model to retain only a subset of the matches. Our experiments indicate that this partial selection is already sufficient to maintain strong performance on long sequences. We further elaborate on this discussion in Q5. We thank the reviewer for the insightful suggestion, which helped strengthen the evidence supporting our analysis of token collapse.
>
> **Table D: Average cosine similarity of global attention over blocks.**
>
> | Block      | 0      | 1      | 2      | 3      | 4      | 5      | 6      | 7      | 8      | 9      | 10     | 11     | 12     | 13     | 14      | 15     | 16     | 17     | 18     | 19     | 20     | 21     | 22     | 23     |
> | ---------- | ------ | ------ | ------ | ------ | ------ | ------ | ------ | ------ | ------ | ------ | ------ | ------ | ------ | ------ | ------- | ------ | ------ | ------ | ------ | ------ | ------ | ------ | ------ | ------ |
> | Similarity | 0.7031 | 0.1824 | 0.9674 | 0.9785 | 0.9909 | 0.9805 | 0.9857 | 0.9870 | 0.9707 | 0.9733 | 0.9303 | 0.6608 | 0.5388 | 0.0809 | -0.0563 | 0.3196 | 0.9401 | 0.9167 | 0.9915 | 0.9928 | 0.9889 | 0.9043 | 0.8958 | 0.9707 |
>
> **Q5: Discussion on mitigating error accumulation
> A5:** Thank you for the insightful hypothesis. We agree that merging early-stage redundancy encourages VGGT to form a more compact and stable global representation, which in turn reduces unnecessary computation in later layers. Beyond efficiency, we believe this design is also related to mitigating error accumulation along the depth of the network.
> First, VGGT fundamentally learns higher-level keypoint correspondences through global attention, as evidenced by our attention-map visualizations, where high activation values consistently concentrate around keypoint-like regions. In such a setting, redundant or noisy tokens can repeatedly participate in attention operations and propagate their errors to deeper layers, especially in long sequences. By aggressively merging redundant tokens at early stages, our method effectively reduces the number of unstable or low-saliency keypoints that enter subsequent attention blocks, which acts as a form of regularization and helps stabilize the representation as depth increases.
>
> Second, there is a conceptual parallel to classical SLAM and SfM pipelines, where sparse but reliable keypoints are often sufficient to achieve high-quality pose estimation. Prior works have shown that simply increasing the number of matched keypoints does not necessarily lead to better performance; in some cases, it can even degrade estimation quality due to the introduction of more outliers and noisy matches \[1]. From this perspective, our token merging can be viewed as automatically pruning redundant or less informative keypoints, allowing the model to operate on a more appropriate number of informative tokens, especially in long video sequences.
> Taken together, these observations suggest that early token merging is not only an efficiency mechanism but also a strategy to control error propagation: it filters out redundancy and potential noise before they are repeatedly amplified by deep stacks of attention layers. We have added this discussion to the revised version to better articulate the connection between our design and the mitigation of error accumulation.
>
> **Summary**
> We hope our response can address all of your concerns raised in the rebuttal. If the reviewer still has any remaining questions or concerns, we would be glad to hear that.
>
> **Reference**
>
> \[1] Good feature matching: Toward accurate, robust vo/vslam with low latency. In TPAMI 2021.

---

> ### Author Response · Authors · 2025-11-27
> **Thanks for your recognition of the value of this work, we sincerely look forward to your feedback.**
>
> Dear Reviewer SPNW,
>
>
> Thank you once again for your valuable comments on our submission. As the discussion phase is approaching its end, we would like to kindly confirm whether we have sufficiently addressed all of your concerns (or at least part of them). Should there be any remaining questions or areas requiring further clarification, please do not hesitate to let us know. If you are satisfied with our responses, we would greatly appreciate your consideration in adjusting the evaluation scores accordingly.
>
>
> We sincerely look forward to your feedback.

---

> ### Comment · Reviewer_SPNW · 2025-11-28
>
> I would like to thank the authors for their comprehensive response and the significant effort put into conducting additional experiments during the rebuttal period.
>
> One minor follow-up regarding the intriguing robustness results in Response A3 (Table C): The fact that FastVGGT outperforms the Baseline even more significantly when the first frame is corrupted (occluded/blurred) is a very strong result. Could you elaborate slightly on the mechanism here? Does this imply that your token merging process effectively acts as an 'implicit outlier rejection' mechanism for the reference frame? (i.e., corrupted tokens in the first frame likely have low similarity with valid tokens in subsequent frames, so they are perhaps naturally 'filtered out' or their influence is diluted during the merging process, whereas the Baseline model is forced to attend to these noisy reference tokens fully?) Adding a brief intuition about this 'implicit cleaning' effect to the final discussion would further strengthen the paper's narrative."
>
> The authors have satisfactorily addressed most of my concerns. The additional experiments and analyses have significantly strengthened the paper, particularly regarding the robustness and the justification for the design choices.
>
> I am inclined to raise my score to reflect the improved quality and completeness of the manuscript. I strongly encourage the authors to incorporate these new results and the associated discussions into the final version of the paper.

---

> > ### Author Response · Authors · 2025-11-28
> > **Further Discussion on First-Frame Corruption**
> >
> > Thank you for your valuable suggestions and recognition. Following your advice, we further reflected on the robustness exhibited by FastVGGT under the condition of first-frame corruption, and have added a supplementary discussion in Appendix Section D.
> >
> > We agree with your insightful point that corrupted tokens in the first frame are diluted during the merging process. In addition, we would like to offer the following perspective: Analogous to traditional SLAM systems, where the first frame is typically used for keypoint initialization, corruption in this frame can compromise both the quantity and reliability of the extracted keypoints. When these degraded keypoints are matched against highly redundant ones from subsequent frames, the likelihood of mismatches and noise is significantly increased. In contrast, FastVGGT’s merging strategy mitigates such issues by reducing redundancy, thereby helping to suppress the propagation of noise introduced by early-frame corruption.
> >
> > We sincerely appreciate your thoughtful reading and the time you devoted to reviewing our work. We would be pleased to provide further clarifications or additional analyses if the reviewers have any remaining questions.

---

### Author Response · Authors · 2025-11-24
**Summary of Author Response to All the Reviewers**

We sincerely thank all reviewers for their time and constructive feedback.

In response to the comments, we have revised several sentences throughout the paper to improve clarity, with all changes highlighted in blue in the revised paper. We summarize the main experimental additions and revisions below:

1.  Added complete details of all experimental settings, including the testing datasets, merging ratio, and input image resolution in Section 4.1.
2.  Added a comprehensive quantitative comparison against Efficient-Transformer baselines (ToMe-random, ToMe-stride, PiToMe) in Section 3.1 (please see Table E in this comment).
3.  Added a quantitative experiment on short sequences with different merging ratio in Appendix C (please see Table A in this comment).
4.  Added a quantitative comparison of TopK and fix-stride strategy for salient token selection in Section 4.4 (Table B in this comment).
5.  Added both quantitative and qualitative experiments with blurred and occluded first frames to demonstrate robustness, as shown in Appendix D (please see Table C in this comment).
6.  Added a quantitative analysis of token collapse in Section 4.5 (please see Table D in this comment).
7.  Added a qualitative comparison of GPU memory usage between VGGT and VGGT\* in Section 3.4 (please see Table D in this comment).
8.  We updated all result tables to avoid ambiguity by marking the top three results with colors and bolding the best one.
9.  Added qualitative experiment to failure case Appendix C.

We would be pleased to provide further clarifications or additional analyses if the reviewers have any remaining questions.

---

### Author Response · Authors · 2025-12-01
**Summary of Discussion to AC**

Dear Area Chair and Senior Area Chair,

We are aware of the recent incident of OpenReview and the corresponding measures implemented by the ICLR Program Chairs (including score reverts and new AC assignments). We fully understand and appreciate the efforts made by the organizers, reviewers, and you to uphold the integrity of the review process.

We wanted to bring to your attention that prior to the incident, we had already submitted comprehensive responses to all reviewers, summarized in the General Response. We believe that these responses addressed most of the concerns raised during the review phase. Importantly, **Reviewer rRZP increased their score to 8, and Reviewers SPNW and aGnr confirmed that their concerns had been fully addressed and expressed support for the acceptance of our paper. However, due to the timing of the leakage incident, they were unable to update their scores in the system.** With regard to Reviewer f7WY, we believe that we have adequately addressed their concerns. Unfortunately, they were not able to submit a follow-up response before the discussion period ended.

We sincerely appreciate the time, diligence, and effort you have invested in the reviewing process. We hope that these clarifications are helpful for your decision-making. Thank you very much for your careful consideration of our submission.

Sincerely,



The Authors of Paper 5717

---

### Meta-Review · Area_Chair_Hmed · 2026-01-15

**Summary:**

**Reviewer SPNW:** Insufficient discussion of performance trade-off on short sequences (W1)(revised, Appendix C). Weak relation between motivation and implementation for "salient token" selection (W2, W5)(Appendix F.2). Robustness to "reference frame" quality is questionable (W3, Q3)(revised, Appendix D). Lack of quantitative evidence for "token collapse" (W4, Q4)(revised, Section 4.5). Mechanism for error accumulation mitigation by removing tokens is not discussed (Q5)(commented, not revised).

**Reviewer aGnr:** Insufficient experimental details / lack of comparisons (1)(revised, Section 4.1). Inconsistent conditions in baseline comparison (2)(revised, Table 1). Limited technical contribution (3)(revised, Sections 3.1, 4.5).

**Reviewer f7WY:** Limited improvement (W1)(revised). Memory consumption not analyzed (W2)(revised, Figure 7). Limited comparison (W3)(claimed that $\pi^3$ already avoids memory redundancy). Unclear meaning of bold numbers in tables (Q1)(Tables revised). Unclear advantage in handling long sequences (Q2)(Section 5).

**Reviewer rRZP:** Incremental contribution (W1). Positioning of the work within the existing body of work is not good (W2, W3, Q2)(revised, Sections 2.2, 3.1). Limited discussion on failures / limitations (W4, Q4)(revised, Appendix C). Unclear meaning of "geometry-aware" (Q1)(commented). The merging ratio as a fixed hyperparameer (Q3)(commented).

**Additional points:**
- Line 269 the Chamfer Distance (CD),
- Line 311: top-K → top-k
- Line 342: introduce a(n) simple
- Line 362: L=24 → $L=24$
- Table 5: It would be more descriptive to write "VGGT*" and "FastVGGT", rather than "Baseline" and "Ours".

**Reviewer Concerns:**

Although the revision made on this manuscript has been extensive, I think that as a result of the revision most of the reviewer concerns have been addressed appropriately.

**Reviewer Scores:**

The initial evaluation of Reviewer rRZP was already on the positive side, and mentioned that he/she will further increase his/her score. Although Reviewers SPNW and aGnr initially evaluated this paper negatively, they also mentioned increasing their scores after the author responses and revision. The initial evaluation of Reviewer f7WY was negative, but I think that most of his/her concerns have been addressed in the revision to a certain extent.

---

### Decision · Program_Chairs · 2026-01-26

Accept (Poster)